# Drone Measurements of Surface-Based Winter Temperature Inversions in the High Arctic at Eureka

Alexey B. Tikhomirov[1,*], Glen Lesins[1], and James R. Drummond[1]

[1]Department of Physics and Atmospheric Science, Dalhousie University, PO Box 15000, Halifax, Nova Scotia, B3H 4R2, Canada

**Correspondence:** Alexey B. Tikhomirov (alexey.tikhomirov@dal.ca)

**Abstract.** The absence of sunlight during the winter in the High Arctic results in a strong surface-based atmospheric temperature inversion especially during clear skies and light surface wind conditions. The inversion suppresses turbulent heat transfer between the ground and the boundary layer. As a result, the difference between the surface air temperature, measured at a height of 2 m, and the ground skin temperature can exceed several degrees Celsius. Such inversions occur very frequently in polar regions and are of interest to understand the mechanisms responsible for surface-atmosphere heat, mass and momentum exchanges and are critical for satellite validation studies.

In this paper we present the results of operations of two commercial remotely piloted aircraft systems, or drones, at the Polar Environment Atmospheric Research Laboratory, Eureka, Nunavut, Canada, at 80°N latitude. The drones are the Matrice 100 and Matrice 210 RTK quad-copters manufactured by DJI and were flown over Eureka during the February-March field campaigns in 2017 and 2020. They were equipped with a temperature measurement system built on a Raspberry Pi single-board computer, three platinum wire temperature sensors, Global Navigation Satellite System receiver, and a barometric altimeter.

We demonstrate that the drones can be effectively used in the extremely challenging High Arctic conditions to measure vertical temperature profiles up to 75 m above the ground and sea ice surface at ambient temperatures down to -46 °C. Our results indicate that the inversion lapse rates within 0-10 m altitude range above the ground can reach the values of ∼10-30 °C/100 m (∼100-300 °C/km). The results are in a good agreement with the coincident surface air temperatures measured at 2, 6 and 10 m levels at the National Oceanic and Atmospheric Administration flux tower at the Polar Environment Atmospheric Research Laboratory. Above 10 m more gradual inversion with an order of magnitude smaller lapse rates is recorded by the drone. This inversion lapse rate agrees well with the results obtained from the radiosonde temperature measurements. Above the sea ice drone temperature profiles are found to have an isothermal layer above a surface based layer of instability which is attributed to the heat flux through the sea ice. With the drones we were able to evaluate the influence of local topography on the surface-based inversion structure above the ground and to measure extremely cold temperatures of air that can pool in topographic depressions. The unique technical challenges of conducting drone campaigns in the winter High Arctic are highlighted in the paper.

# 1 Introduction

Atmospheric temperature is one of the key parameters used to study climate (World Meteorological Organization, 2021). Atmospheric temperature measurements are conducted in-situ using different types of temperature sensors installed at the meteorological observing stations on the ground (Taalas, 2018), marine platforms, i.e. ships and buoys (Cold Regions Research and Engineering Laboratory; International Arctic Buoy Programme; National Data Buoy Center; Multidisciplinary Drifting Observatory for the Study of Arctic Climate), and airborne platforms, i.e. radiosonde (Luers and Eskridge, 1998; DuBois et al.,

2002), dropsonde (Skony et al., 1994; Cohn et al., 2013; Wang et al., 2013; Intrieri et al., 2014), sounding rocket (Webb et al., 1961) and aircraft (Antokhin et al., 2012; McBeath, 2014; Nédélec et al., 2015; Berkes et al., 2017). In-situ measurements provide high accuracy and high temporal resolution temperature datasets and serve as a "golden standard" for validation for other methods.

Atmospheric air temperatures are also derived from the measurements conducted using remote sensing instruments, i.e.

radiometers (Tomlinson et al., 2011; Pietroni et al., 2014) and LIDARs (Behrendt, 2005) installed on the ground, airborne and satellite-borne platforms. Satellite temperature observations are especially valuable, since they provide global coverage reaching the areas where ground-based air temperature measurements are challenging due to a small number of monitoring sites, i.e. above ocean surface (Jackson and Wick, 2010), in mountain ridges (Orellana-Samaniego et al., 2021), in the Arctic and Antarctic (Soliman et al., 2012).

The World Meteorological Organization (WMO) assesses global temperature fields and temperature anomalies based on the measurements of air temperature at 1.25 to 2 m above the ground level on the land. These temperatures are referred to "surface" or "near-surface" air temperatures (SATs, Rennie et al., 2014). SATs measured by means remote sensing are also used in the WMO assessments. However, they are derived from so called "skin" temperatures, which are temperatures at the surface-air interface, since they are retrieved from radiometric measurements (Li et al., 2013). Also, satellite-based temperature

datasets suffer from missing data due cloud interference. A review of current cloud filtering approaches and a novel method for recovering the temperatures under cloudy skies can be found in Wang et al. (2019) and references therein.

In polar regions in the absence of sunlight strong surface-based temperature inversions (SBIs) occur frequently. The occurrence rate is >70 and >90% of the time in the Arctic and Antarctic respectively during winter months (see Bradley et al. (1993); Walden et al. (1996); Hudson and Brandt (2005) and references therein). Due to terrestrial radiative cooling of the

surface in clear sky conditions and suppressed turbulent heat transfer between the ground and the boundary layer under light wind conditions, the difference between SAT and skin temperature can be significant. Based on temperature measurements conducted at the South Pole in the winter of 2001, it was found that "median difference between the temperatures at 2 m and the surface" could reach 1.3 °C in winter in clear sky conditions, which is equal to a 65 °C/100 m (650 °C/km) inversion lapse rate (Hudson and Brandt, 2005). According to the observations the strongest temperature gradient is confined within a

0.2 m air layer above the surface where the temperature difference is equal to 0.8 °C leading to a 400 °C/100 m (4000 °C/km) inversion lapse rate. This difference between 2 m SAT and skin temperature results in a negative bias in the SAT products obtained from the satellite measurements (see Adolph et al., 2018 and references therein). Between 2 m and 100 m the monthly

mean temperature gradient varies within 11.1-12.8 °C/100 m (111-128 °C/km) in March-September as reported by Hudson and Brandt (2005) based on radiosonde (RS) data from South Pole covering 1994–2003.

Pietroni et al. (2014) studied the characteristics of SBI over the course of a year of 2005 at Dome C, Antarctica. They measured temperature profiles up to 205 m from the ground using a scanning microwave radiometer with 10-50 m vertical resolution. SBI temperature gradients during the summer months was found to be between 0 and 15 °C/100 m (0-150 °C/km). SBI was observed in 67% of the time and some daily cycle was registered in temperature profiles. During the winter months the SBI temperature gradient could exceed 30 °C/100 m (300 °C/km). The SBI was observed in 99% of the time with no diurnal
cycle in the temperature profiles.

    Boylan et al. (2016) characterised SBIs over Antarctica based on the data from satellite observations (Infrared Atmospheric Sounding Interferometer, IASI), atmospheric reanalysis model (ERA-Interim) and dropsondes. They found that over land IASI SATs, derived by interpolating temperatures at the lowest retrieved level, are 3.10 °C larger than dropsonde SATs. Over sea ice, in contrast, IASI SATs are found to be 3.45 °C smaller than dropsonde SATs. These differences are associated with extremely
shallow inversion layers that satellite products can not resolve. Due to that accurate satellite-based SBI measurements are limited to relatively deep SBIs.

    According to Bradley et al. (1993) the mean rate of the temperature change within the inversion measured in December-March during 1967-1986 at the RS sites in the Canadian Arctic and Alaska is in the range of 1.5-2.3 °C/100 m (15-23 °C/km), which is much less than was measured in Antarctica. Walden et al. (1996) reported the multi-year monthly averaged inversion
lapse rate for 0-250 m altitude as being between 1.2 and 1.8 °C/100 m (12-18 °C/km) in Barrow, Alaska (1953-1990) and between 2.0 and 2.4 °C/100 m (20-24 °C/km) in Eureka, Nunavut (1967-1990). Lesins et al. (2010) reported on a weakening of the winter inversion strength at Eureka from 1985 to 2007 using the station RS observations. Zhang et al. (2011) analysed a dataset covering 20 years (1990–2009) of RS observations from 39 Arctic and 6 Antarctic sites and compared it to a reanalysis dataset and to simulations from climate models to examine spatial and temporal variability of SBI including frequency of
occurrence, depth and intensity and relationships among them. They found the strength, occurrence frequency and depth of the SBI are larger in winter and fall than in summer and spring and are positively correlated between each other, both spatially and temporally. Also all three characteristics are in inverse relationship with surface temperature. Lesins et al. (2012), based on the data from 22 Canadian Arctic RS stations covering 1971 to 2010, suggested that a strong SBI plays an important role in Arctic amplification of climate change. Smith and Bonnaventure (2017) analysed air and ground temperature data
collected at Alert, Nunavut, Canada and found the SBI occurrence may have an effect on the spatial variation in the High Arctic permafrost thermal state, specifically in the regions with thin snow cover. Pavelsky et al. (2011) showed a correlation between the inversion strength and annual sea ice concentration in the Arctic and Antarctic. After analysing data on the near-surface temperature inversions from the Atmospheric Infrared Sounder they suggested the inversion strength could be controlled by the ice concentration through modulation of the surface heat fluxes. Thus, monitoring and characterisation of SBI remains
important in understanding its role in atmospheric processes and ocean-atmosphere interaction.

    In recent years Remotely Piloted Aircraft Systems (RPAS), or drones, have become a commonplace in industry and science (Cassano, 2014; Chabot and Bird, 2015; Kräuchi and Philipona, 2016; Cowley et al., 2017; Kral et al., 2018; Mašić et al., 2019;

Barbieri et al., 2019; Gaffey and Bhardwaj, 2020; Lampert et al., 2020a,b; de Boer et al., 2020; Wenta et al., 2021; Varentsov et al., 2021). There are two main drone types used in research: fixed-wing and rotary-wing. Both have certain advantages and

95 limitations which affect the performance in specific situations (González-Jorge et al., 2017).

When two RPAS of the same mass are compared to each other, fixed-wing RPAS usually outperform rotary-wing drones in terms of flight endurance, design simplicity and cost of operation and maintenance. Fixed-wing RPAS are optimal for large area surveys where longer endurance, faster speed and hence larger spatial coverage are the most critical factors (Jouvet et al., 2019; Zappa et al., 2020). For these reasons they are widely used for meteorological and atmospheric science applications

(Knuth and Cassano, 2014; Cassano, 2014; Cassano et al., 2016; de Boer et al., 2018; Bärfuss et al., 2018; Zappa et al., 2020) and glaciology (Jouvet et al., 2019). On the other hand, rotary-wing drones have better payload capacity and can hover in one spot, which is critical for photography surveys. Their lower speeds and superior manoeuvrability means that flying in the rough topographic environment become less challenging (Shahmoradi et al., 2020). Rotary-wing RPASs are easier to operate and they do not require special equipment such as a catapult or a long runway for a launch.

Research projects which require to cover small areas (a few km$^2$), carry simple and lightweight payloads (a few kg) and fly within hundred meters above the ground with complex topography became feasible due to remote operation capabilities, high mobility and manoeuvrability and low operational costs of the drone in comparison with manned aircraft. Easier access to the drone pilot training programs, shorter amount of time required for learning and certification as well as piloting independence make the drones attractive for small scale research initiatives.

Advantages and limitations of a small fixed-wing airborne measurement system (DataHawk, 1 m wingspan, 0.7 kg take-off weight) for in-situ atmospheric measurements within and above the boundary layer are discussed by Lawrence and Balsley (2013). The DataHawk is built around an "off-the-shelf" elastic foam airframe and a low-cost custom-designed autopilot. It is equipped with a suite of sensors to measure temperature, humidity and wind at ∼1 m spatial resolution, >1 km horizontal scale and within the altitude range from a few meters up to 9 km (balloon-drop deployment option) above the ground level.

The application of multi-rotor RPAS to study temperature inversions up to 1000 m above the ground in urban areas in winter time has been reported by Mašić et al. (2019). Their drone was built based on an open-source flight controller and commercially available propulsion system and carbon fibre frame. These authors conducted observations in many different SBI scenarios at ambient temperatures falling below -20 °C in the context of local air pollution. They found a correlation between the hazardous air pollution events and strong and shallow SBIs formed below 150 m altitudes above the ground lever.

During such conditions the SBI temperature gradients could reach values larger than 3 °C/100 m (30 °C/km). It was also noticed that at 2-3 m/s drone vertical speeds, temperature profiles are affected by a response time of the temperature sensor. This resulted in a hysteresis patterns in the temperature vertical profiles measured on the ascent and descent similar to those previously reported by Cassano (2014) and cannot be neglected. The hysteresis was corrected by calculating the arithmetic mean of the temperatures recorded at each altitude level during the ascending and the descending phases. Mašić et al. (2019)

also highlighted the advantages of using drones in comparison with other techniques (RS, microwave radiometry, cable-car and ground-based measurements). Among those advantages are lower operation cost per single temperature profile, good control

over the flight parameters, ability to measure temperature during both ascent and descent and finer vertical resolution. All of them are critical for SBI measurements.

In polar regions drones provide unique opportunities to conduct studies in rapidly changing and often hard to predict environ-
mental conditions due to low risks of operation. However, the harsh environment of high latitudes including temperatures below -30 °C, proximity to the Earth's magnetic pole, poor performance of Global Navigation Satellite System (GNSS) receivers and complete darkness during the polar night pose challenges for drone operations (Gustafsson and Bendz, 2018).

A review of research applications of a smaller (up to ∼25 kg) and larger (∼500 kg) fixed-wing RPAS in Antarctica between 2007 and 2013 together with the results of observations of atmospheric boundary layer temperatures using the Small Unmanned
Meteorological Observer (SUMO) drone (Reuder et al., 2012) in the vicinity of McMurdo Station is presented by Cassano (2014). SUMO is designed around "off-the-shelf" expanded propylene airframe and an opensource autopilot system (Reuder et al., 2009). Observations made during SUMO drone flights covered various meteorological situations including well-mixed and stable boundary layers as well as the situations where the boundary layer was rapidly changing. The flights were conducted within -29 and 0 °C temperature range and up to 1400 m altitude above the ground level. Cassano (2014) concluded that the
SUMO drone proved itself to be an effective tool to measure sharp, shallow SBIs due to its small dimensions and light weight (0.80 m wingspan, 0.6 kg take-off weight), deployment and operation simplicity and low cost. However, it was pointed out that short (∼30 min) endurance of the drone limited the useful maximum range to 5-10 km from the launch/landing site. It was also found that the temperature sensors suffered from a 2.5-5 s time lag, which had to be corrected during data possessing by introducing an appropriate time delay between altitude and temperature readings (Mahesh et al., 1997).

Technical difficulties and examples of application of a 19 kg quad-copter custom built for polar missions to study atmospheric boundary layer at 79°N in Greenland and deployable from a research vessel has been recently reported by Lampert et al. (2020a). These authors measured vertical profiles of meteorological parameters within 1000 m altitude range at up to 8 m/s drone ascent/descent speeds and provided detailed analyses of their findings and factors affecting the results such as the impact of rotor blades, turbulent fluctuations and heat produced by drone motors on temperature measurements. They suggested a
novel approach for time lag correction in which the temperature sensor response time is not fixed, but is tied to the vertical velocity to handle changing directions and rates of air flow around the sensors. They also highlighted that due to Earth's magnetic field anomalies and magnetic disturbances produced by the vessel, takeoff and landing had to be performed manually and certain adjustments had to be applied to the autopilot system to correct for this during operations.

Many research groups have developed RPASs on open-source platforms (Ebeid et al., 2017) and optimized them for specific
applications (Roldán et al., 2015; Kräuchi and Philipona, 2016; Villa et al., 2016; Jouvet et al., 2019; Lampert et al., 2020b). Others have utilized "off-the-shelf" airframes (Cowley et al., 2017; Burgués and Marco, 2020; Varentsov et al., 2021) and their modifications (Reuder et al., 2012; Lawrence and Balsley, 2013; Mašić et al., 2019; Segales et al., 2020) or sophisticated commercial solutions (Knuth and Cassano, 2014; Cassano et al., 2016; Bärfuss et al., 2018; Zappa et al., 2020). Both approaches have their merits and a variety of successful examples can be found in the literature (Gaffey and Bhardwaj, 2020). However,
many factors like technology availability and flexibility, equipment and maintenance cost have to be taken into consideration during project planning.

As drone technology emerged and became more accessible recently, we started to develop a concept to study SBI at the Polar Environment Atmospheric Research Laboratory (PEARL, Fogal et al., 2013) in Eureka, Nunavut, Canada, at 80°N latitude with RPAS in 2016. We were driven by the idea of using a commercial "turn-key" drone solution for our application. Keeping this

in mind, the plan was to evaluate and learn whether an "off-the-shelf" rotary-wing drone can be economic, robust and reliable in the High Arctic environment, so the time and efforts spent on the development of a custom system can be saved.

The goal of this paper is to present the results of the first pilot studies of the temperature profiles within 75 m of the ground conducted in Eureka using a custom built temperature sensing system installed on a commercial rotary-wing RPAS.

To achieve the goal the following tasks have been accomplished.

Technical tasks:

- Two commercial quad-copters with different navigation systems were identified, acquired and flown in Eureka to demonstrate and evaluate the feasibility of conducting drone operations at 80°N (see subsection 2.1).

- A custom built temperature measurement system was installed and tested onboard the quad-copters to evaluate its potential in providing reliable air temperature data (see subsection 2.2).

- The quality of air temperature measurements, conducted in field conditions, relative to sensor locations onboard the drone was examined using three identical temperature sensors (see subsections 3.1.2 and 3.2.2).

Scientific tasks:

- The results of the drone SBI measurements were validated against the data from the flux tower, radiosondes and weather stations in Eureka (see subsections 3.1.2 and 3.2.2).

- Drone vertical temperature profiles collected over flat terrain and in a gully in Eureka were examined to determine the role of local topography on SBI shaping (see subsection 3.2.3).

- Drone vertical temperature profiles collected over the sea ice were examined for the signs of the heat flux through the sea ice (see subsection 3.2.4).

The paper describes the results of the tests and measurements, discusses the performance of the drones and the challenges

of conducting drone operations in the High Arctic in winter conditions.

## 2    Methods and instrumentation

### 2.1    *Remotely piloted aircraft systems*

Two RPASs have been identified, acquired and tested to study SBI in the harsh environment of the High Arctic in Eureka. Both drones are commercially available quad-copters manufactured by DJI. The first drone, Matrice 100 (M100), is a development

grade quad-copter with 650 mm diagonal wheelbase and 3.6 kg maximum takeoff weight. The drone can be powered either from a standard (TB47D, 4500 mAh) or extended (TB48D, 5700 mAh) capacity lithium polymer (LiPo) battery and can be configured to use a single battery or two in parallel. Depending on the configuration typical hovering time can vary between

19 and 40 min with 0.5-1.2 kg of payload. Both TB47D and TB48D batteries are equipped with internal temperature sensors. The readings from the sensors are displayed on the screen of the remote controller tablet during the flight. For navigation the drone relies on its Inertial Measurement Unit (IMU), compass and GNSS. It allows one to conduct flights in so-called positioning mode (P-mode) and attitude mode (A-mode). In P-mode the drone utilizes onboard GNSS receiver and barometric altimeter to maintain its horizontal and vertical position. Bearing information is taken from the onboard compass. According to specification, drone's hovering accuracy in P-mode is better than 0.5 m and 2.5 m in vertical and horizontal directions respectively. In A-mode the drone only utilizes its barometric altimeter to maintain altitude, horizontal position is not retained.

The second drone, DJI Matrice 210 RTK (M210 RTK), is an industrial grade quad-copter. It has 643 mm diagonal wheelbase and 6.14 kg maximum takeoff weight. It employs a pair of standard (TB50, 4280 mAh) or extended (TB55, 7660 mAh) capacity batteries. Both TB50 and TB55 battery types are equipped with internal temperature sensors as well as heaters. The drone telemetry, which includes the temperature of the batteries, is displayed on the screen of the remote controller tablet during the flight. The heater turns itself on if the battery temperature falls below 15 °C to maintain battery's optimal operation conditions. Maximum drone flight time varies between 13 and 32 min depending on the payload weight and type of the batteries installed. Approximate maximum payload is 1.7 kg with a set of standard batteries and 1 kg with extended capacity batteries. The M210 RTK differs from the M100 by its advanced navigation system which employs Real-Time Kinematic (RTK), a differential GNSS technique, which provides high positioning accuracy when used together with a base station in P-mode. According to specification a hovering accuracy of 0.1 m in both vertical and horizontal directions can be reached by utilizing the drone together with DJI D-RTK ground system kit (RTK mode). The drone is also equipped with an obstacle avoidance system to make the flights safer. The air-frames of both drones are made with carbon fibre and aluminum, which makes them suitable for low temperatures. The drones and their payload configuration are shown in Figures 1 and 2. Detailed specifications of the drones can be found on-line: DJI M100 - https://www.dji.com/ca/matrice100, DJI M210 RTK - https://www.dji.com/ca/matrice-200-series. Payload details are discussed further in the paper.

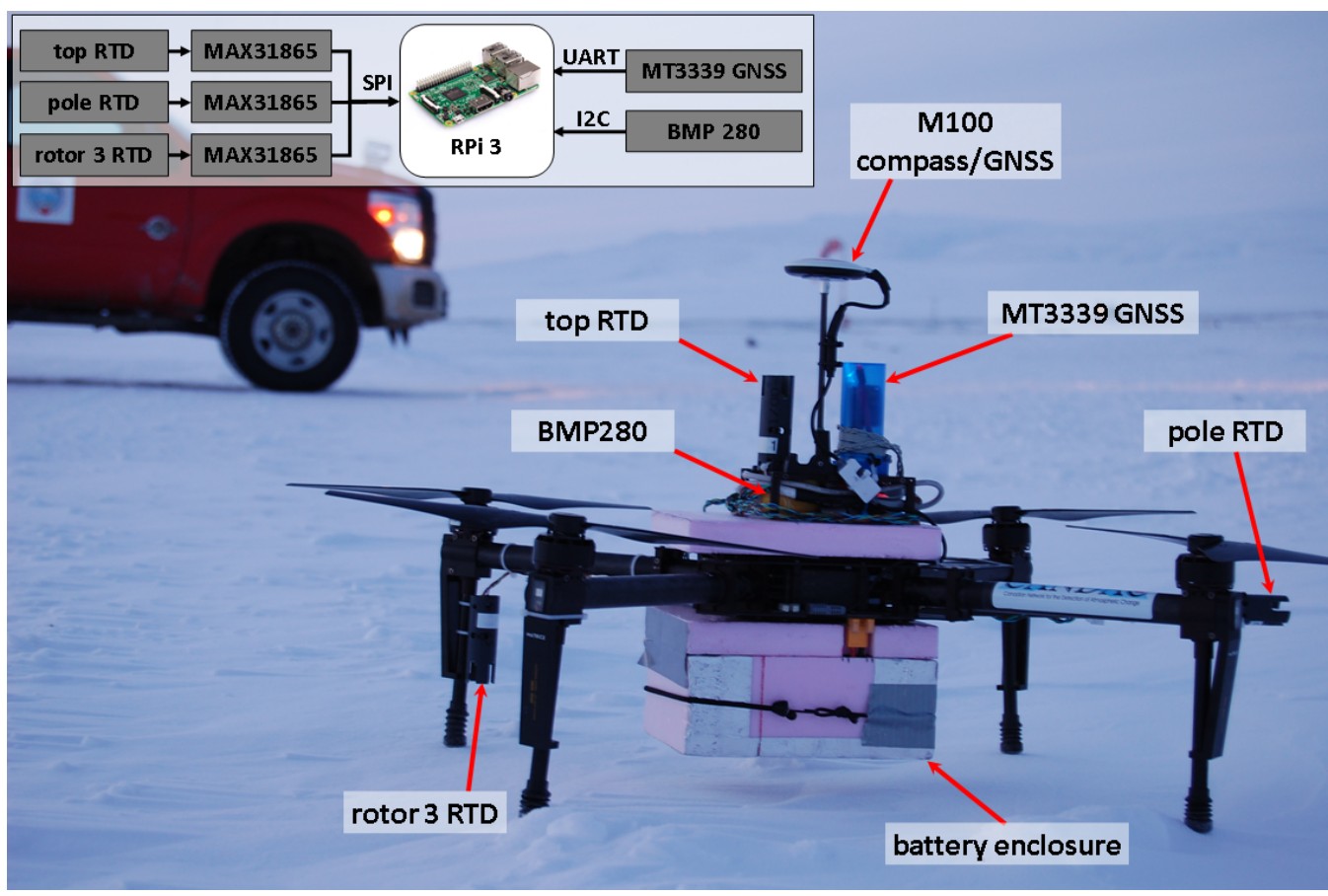

**Figure 1.** DJI M100 drone and its payload.

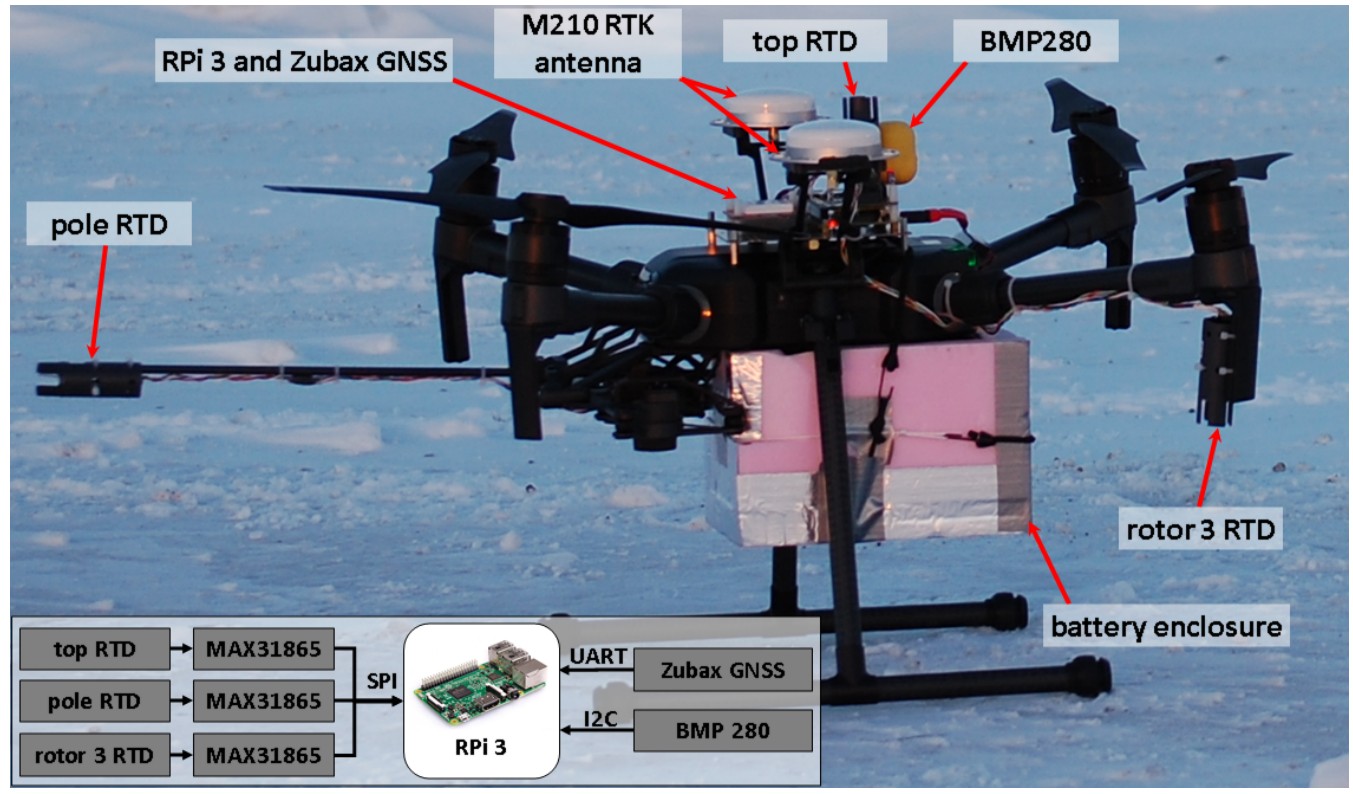

**Figure 2.** DJI M210 RTK drone and its payload.

## 2.2 *Onboard data collection system and sensors*

To record the ambient air temperature during the flights three identical platinum wire Resistance Temperature Detector (RTD) sensors are installed aboard the drones. The RTD sensors are 1 mm in diameter and 15 mm long ceramic wire-wound elements (1PT100KN1510, Omega Engineering, Inc.: tolerance class B, ±0.3 °C at 0 °C). Each RTD element is connected to its own MAX31865PMB1 peripheral module (Maxim Integrated Products, Inc., 2014). The modules employ the MAX31865 resistance-to-digital converter optimized for platinum RTDs. The converter has 0.03 °C resolution and 0.5 °C (0.05% of full scale) total accuracy at 21 ms conversion time according to the datasheet. The modules with RTD elements are housed in a 25 mm diameter and 75 mm long PVC tubes for protection. The first module is attached to the top side of the drone close to its center point (top RTD, see Figures 1 and 2). The second module is attached to the tip of a ∼60 cm long pole at the front side of the drone (pole RTD). This is done to minimize the influence of turbulent air flows produced by the drone's propellers on temperature measurements (Greene et al., 2018; Lampert et al., 2020a). The third module is mounted under the left rear rotor (rotor 3 RTD). All three modules are connected to the onboard data collection system via a Serial Peripheral Interface (SPI).

To record the horizontal location of the drone a spare GNSS module is installed onboard. The module utilizes MediaTek Chipset MT3339 capable of up to 10 Hz data update rate. The module is connected to the onboard data collection system via UART interface.

To have an altitude reference a separate barometric altimeter is installed. The altimeter is a BMP280 digital pressure sensor (Bosch Sensortec, 2018). It is connected to the onboard data collection system via Inter-Integrated Circuit (I2C) interface. The accuracy of the altimeter is ±0.5 m. The altimeter was verified by comparing its pressure readings to simultaneous measurements taken with a Vaisala WXT-520 weather transmitter within the pressure range between 925 and 1002 hPa. The results showed good agreement between the two sensors (number of data points N = 14791, Pearson's correlation coefficient R = 0.99999).

The onboard data collection system is built on a Raspberry Pi (RPi) model 3 single-board computer with Raspbian operating system (The Raspberry Pi Foundation). Power to the RPi is provided from the drone's extended power port (output voltage range: 18-26V) via 5V universal battery eliminator circuit (UBEC) DC/DC step-down voltage converter. The acquisition code, written in Python, polls each sensor at a sampling frequency of 10 Hz and saves the acquired data in ASCII format to the RPi's microSD card for post processing. Total weight of the data collection system together with the sensors is <0.3 kg.

## 2.3 *Site description*

Drone flights were conducted in Eureka, a small research base located on Ellesmere Island, Nunavut, Canada. The base consists of three main areas: the Environment and Climate Change Canada (ECCC) Weather Station (WS) - a facility complex built at the northern side of Slidre Fjord, on Fosheim Peninsula of the island (79.9890°N, 85.9386°W, 10 m a.s.l., #1 pin in Figure 3), the Eureka Aerodrome (ICAO code: CYEU, 79.9944°N 85.8119°W, 83 m a.s.l., #1 pin in Figure 4) located ∼2.5 km east-north-east of the ECCC WS, and PEARL - an atmospheric research facility, which is operated by the Canadian Network for the

Detection of Atmospheric Change (CANDAC - https://www.candac.ca) and includes several laboratories at different locations within the vicinity of Eureka.

The region around Eureka is a polar desert with mean annual SAT about -19 °C and annual water equivalent precipitation of 70 millimeters (Bernard-Grand'Maison and Pollard, 2018). The region has very little snow cover during the winter period, i.e. 20-30 cm deep snow drifts in the hollows and almost no snow on small hummocks.

In Eureka the standardized meteorological observations are conducted at two observing stations. The first station is a WMO certified site (Eureka Climate or Eureka C, WMO ID: 71613) located ∼100 m east by north of the ECCC WS main building (marked by #2 pin in Figure 3). The second station is a NAV Canada meteorological station (Eureka Aerodrome or Eureka A, WMO ID: 71917) located nearby the Eureka Aerodrome runway (marked by #4 pin in Figure 4). It monitors the weather conditions specific to the aerodrome. The stations measure temperature, relative humidity (RH), wind speed and direction and atmospheric pressure in automatic mode. Additionally, ECCC staff conducts hourly weather observations (visibility and weather conditions) from the rooftop deck of the ECCC WS main building (#1 pin in Figure 3), from which the region of the whole aerodrome down the fjord is visible. Due to ∼2.5 km separation between ECCC WS and Eureka A sites the weather conditions and visibility observed at ECCC WS are assigned to Eureka A for aviation purposes. The results of meteorological measurements at both sites are stored in ECCC archives at 1 hour period for the temperature, dew point, RH, wind speed and direction, visibility and pressure and at 15 minutes period for precipitations.

Also, radiosondes are launched routinely twice a day at 11:15 and 23:15 UTC from the ECCC WS hydrogen shed (#3 pin in Figure 4) all year round. Radiosondes provide vertical profiles of pressure, temperature, relative humidity, wind speed and direction from the ground up to 30-35 km.

Additionally, meteorological measurements are conducted at PEARL. An automatic weather transmitter (Vaisala WXT-510) is installed at the Zero Altitude PEARL Auxiliary Laboratory (0PAL, 79.9905°N, 85.9388°W), located ∼160 m north of the ECCC WS main building (pin #4 in Figure 3). The 0PAL weather transmitter provides data on the weather conditions at its location with one-minute resolution.

The National Oceanic and Atmospheric Administration (NOAA) Flux Tower (FT), a 2 m by 2 m wide and 10 m tall tower, is installed approximately 250 m north-north-east of the East end of the aerodrome runway. Geographical coordinates of the FT are: 79.9955°N, 85.7716°W (see pin #7 in Figure 4). The FT is equipped with temperature sensors (at 2, 6 and 10 m levels relative to the FT base), anemometers, precipitation sensors, barometer and other meteorological and scientific instruments. Detailed descriptions of the FT instrumentation suite and related measurements made at the site can be found in Grachev et al., 2018 and references therein. Measurements at the FT are made at 10 Hz sampling rate for the sonic anemometers, 3 Hz - for the aspirated RTD sensors and once per minute for the rest of the sensors.

Drone temperature measurements were performed during multiple flights at two test sites: the Fjord Test Site and the Runway Test Site. The the Fjord Test Site (FTS, marked in green shading in Figure 3) is a 0.5 km by 0.5 km area on the ice of the Slidre Fjord ∼200 m east-south-east of the ECCC WS and Eureka C meteorological station. The fjord is covered by ice between September and July with an ice thickness of about 0.2-0.5 m in October and reaching 2-2.5 m in May according to the ice surveys performed by ECCC staff at the WS (Ice Thickness Program, Canadian Ice Service). The ice is characterized by low

snow drifts on its flat surface and no signs of cracks or leads during the measurement period. This site was chosen to investigate the features of the SBI above the ice covered ocean.

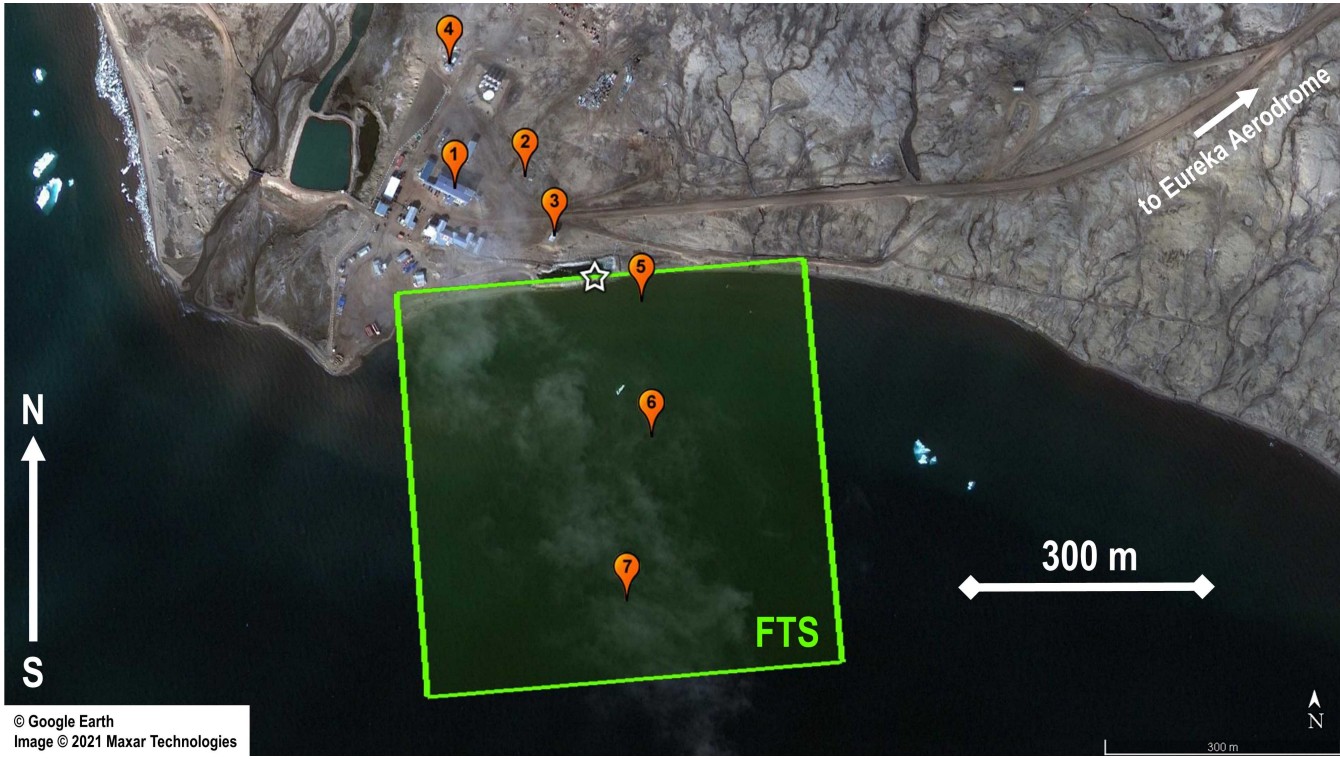

**Figure 3.** The Fjord Test Site (FTS) - a flight region on Slidre Fjord near Eureka Weather Station. Pins in the figure indicate the locations of the ECCC WS main building - #1, Eureka C weather observing site - #2, hydrogen shed (RS launch site) - #3 , 0PAL - #4, temperature profiles measured by M210 RTK in the fjord on 10 March 2020 - #5-#7 (see further details in the text). The drone takeoff/landing pad is marked by a star symbol.

The Runway Test Site (RTS, marked in green shading in Figure 4) is an inverted L-shape area of 1 km by 1 km near the
East end of the runway of Eureka Aerodrome. The site was chosen specifically to study SBI over land due to a favourable
combination of a flat terrain and local topographic features. It is a thermokarst landscape of ice-rich permafrost tundra with
a flat plateau located at the northern side of the runway and surrounded by gullies (Pollard, 2000). The RTS is located in
proximity to the Eureka A meteorological station and includes the NOAA FT. The drone takeoff/landing locations for both
sites are marked by a star symbol in Figures 3 and 4.

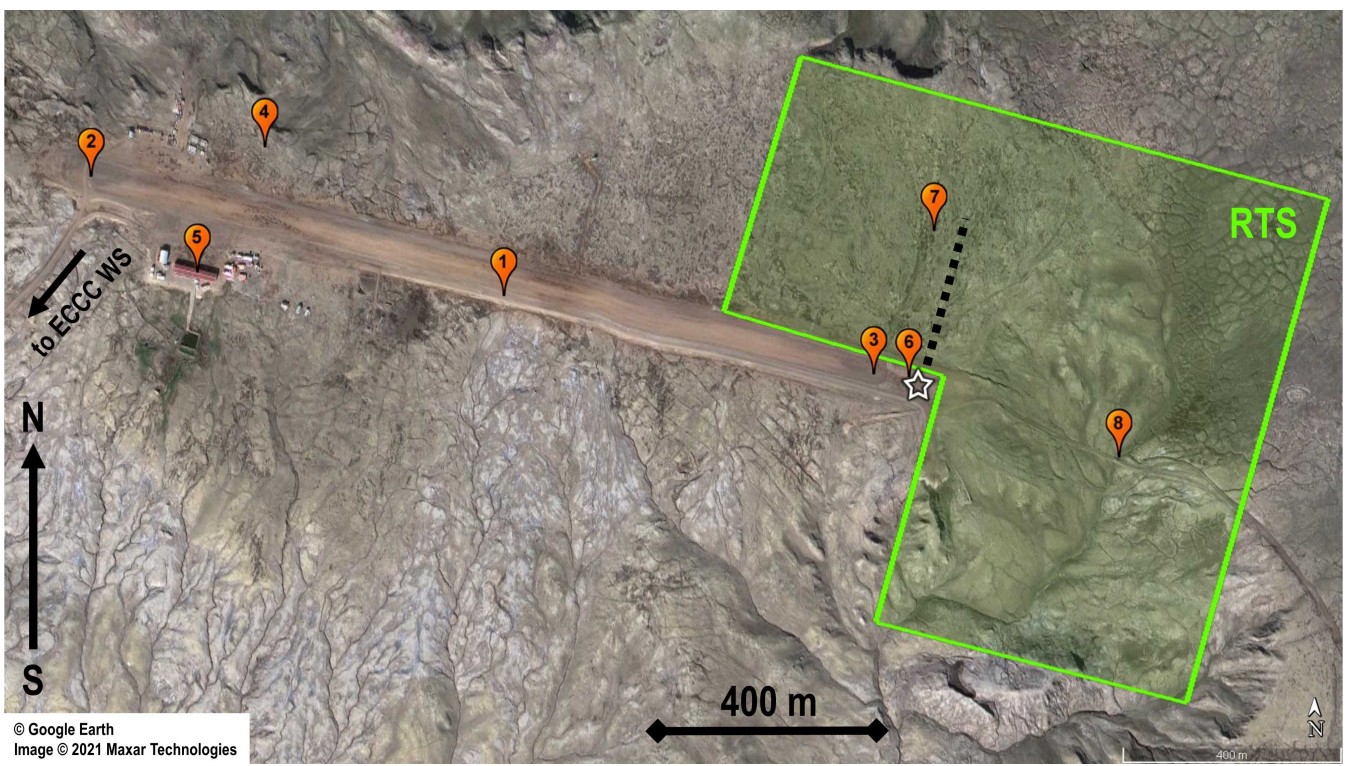

**Figure 4.** The Runway Test Site (RTS) - a flight region near the Eureka Aerodrome and NOAA Flux Tower. Pins in the figure indicate the
locations of the Eureka Aerodrome - #1, west side of the runway - #2, east side of the runway - #3, Eureka A weather observing site - #4,
Fort Eureka buildings - #5, temperature profile measured by M100 on 28 February 2017 - #6, NOAA Flux Tower - #7, gully - #8 (see further
details in the text). Black dotted line represents typical ground track of M210 RTK during the measurements of the vertical temperature
profiles near the Flux Tower in March 2020. The drone takeoff/landing pad is marked by a star symbol.

## 2.4  *Drone batteries in cold environment*

The cold and harsh environment of the High Arctic brings certain challenges to drone operations (see Ader and Axelsson, 2017; Kramar and Maatta, 2018; Lampert et al., 2020a and therein). Among these are very low ambient temperatures, complete darkness during polar night, and navigation difficulties associated with proximity to the the North Magnetic Pole and poor GNSS performance at high latitudes.

According to specifications, certified operation temperatures are: -10 to 40 °C for M100 and -20 to 50 °C for M210 RTK. In the High Arctic typical winter ambient temperatures fall below -30 °C. The main technical challenge associated with cold temperatures is poor performance of lithium batteries (Zhang et al., 2003). While the electronics and mechanics of the drones work well down to -40 °C, the efficiency of the batteries drops drastically below -20 °C, which affects the duration of the flight. According to Pesaran et al., 2013 the optimal range of operating temperatures for lithium batteries spans between 15 and 35 °C. We observed that while the drone's batteries generate internal heat during flight, this cannot keep the batteries within the optimal operation temperature range at below -20 °C ambient air temperatures, even in the case of the M210 RTK which is equipped with battery heaters. To solve this an enclosure made of 25 mm thick extruded polystyrene rigid insulation sheet (R=5 per 25 mm of thickness) was built around the battery compartments of both drones (see Figures 1 and 2). It allows easy installation and removal of the batteries and keeps them at optimal operational temperature during the flight. Battery temperatures were maintained at about 30 °C according to the data from M100 and M210 RTK battery temperature monitoring systems.

## 2.5  *Flight strategy and operation challenges*

All drone operations reported here were performed within the framework of the research activities conducted at PEARL and in accordance with Canadian Aviation Regulations for RPAS. Special procedures were established for operations in the vicinity of Eureka Aerodrome.

The initial flight strategy consisted of several automatic (using an autopilot) or manual flights per day at various locations within the RTS and FTS in the line-of-sight conditions with periodic ascents and descents. Before June 1, 2019, the flights were conducted under Special Operation Flight Certificate, which restricted the maximum flight altitude for the drones to 91 m (300 ft) above the ground level, the minimum visibility - to 4.8 km (3 statute miles) and the minimum celling - to 305 m (1000 ft) above the ground level. After June 1, 2019, the flights were conducted according to the updated Part IX of the Canadian Aviation Regulations, in which the maximum flight altitude for basic operations was extended to 122 m (400 ft) above the ground level. To comply with the updated air space regulations and to increase the number of temperature profiles measured per flight before the drone batteries are drained, in 2020 our maximum flight altitude was ~75 m above the ground level.

Every time before conducting a flight the weather conditions were checked to make sure they are favourable for SBI formation: sky is clear, ambient temperature is below -30 °C, and wind speed is below 5 km/h.

To address the issues associated with effects of cold weather on the performance of the drone pilot all flight controls were conducted from a truck parked nearby the flight region. Also the pilot wore touchscreen friendly electrically heated gloves to be able to navigate the drone using a tablet and to keep their hands warm.

Potential challenges associated with propeller icing and darkness during the operations did not occur. At below -30 °C ambient temperature and at ~70% relative humidity (corresponds to 354 ppmv water vapour mixing ratio) the air was very dry and we did not observe any indications of icing on the propellers nor on the drone airframe during the flights (for comparison, 70% relative humidity at 0 °C corresponds to 4257 ppmv water vapour mixing ratio). Since the operations were conducted at the end of February to the beginning of March there was enough sunlight during the day to perform the flights in well illuminated conditions.

The challenges and solutions related to drone navigation are discussed in subsequent sections of the paper.

## 3   Results and Discussion

### 3.1   *M100 drone*

#### 3.1.1   *M100 first test flights and navigation challenges*

Our first tests with M100 in Eureka (79.99°N, -85.77°W) were conducted in February 2017. The purpose of the tests was to evaluate the possibility of automatic flights and demonstrate the capability of the sensors and data collection system to provide reliable data at ambient temperatures below -30 °C in the High Arctic.

The drone was programmed to perform automatic flights according to a predefined way-point pattern at constant altitude above the ground within the RTS in P-mode (see Figure 4). Unfortunately, in Eureka M100 automatic flights were unsuccessful. The drone failed to maintain constant altitude and systematically climbed up during the course of the flight while the telemetry indicated that the flight was performed at fixed altitude. By the end of each automatic flight the drone could gain an extra 30-50 m of altitude relative to predefined settings. Also there were many cases when the drone lost its bearing and flew in circular patterns.

Similar tests conducted in Halifax, NS, Canada, located at a more southerly latitude (44.6°N, 63.6°W), did not have such problems and the drone performance was satisfactory during those flights. We associate these navigation issues with a failure of M100 navigation system to lock on the GNSS signal and poor performance of the internal compass and barometric altimeter in the High Arctic latitudes and at low temperatures.

Due to this, all further tests with M100 in Eureka were performed in A-mode. Since altitude maintenance was problematic as well, the flights were conducted in true manual mode based on visual observations.

#### 3.1.2   *M100 temperature measurements*

During further flights we tested the performance of the sensors and data collection system. The drone was flown in a pattern with periodic ascents and descents to measure vertical temperature profiles (three profiles total). Each vertical profile consisted

of temperature measurements conducted on a single ascent followed by single descent at a fixed location within the RTS (see Figure 4, #6 pin). An example of raw temperature profile (Flight 1, Profile 2) measured above packed snow at the East end of the runway 250 m south of the FT on 28 February 2017 is shown in Figure 5 (a). The altitude scale is taken relative to the location of the drone takeoff/landing pad which is at the same level as the FT base. Temperature variations measured by FT RTD sensors at 3 Hz sampling rate at 2, 6 and 10 m above the surface during the time frame of the drone ascent and descent (19:39-19:41 UTC) as well as 19:00 and 20:00 UTC Eureka A temperatures are also shown in Figure 5 (a) and (b).

Some difference is observed in the temperature profiles carried out on the drone's ascent and descent. This is associated with the response time of the temperature sensors, vertical speed of the drone and air mixing produced by the drone propellers. During the tests the ascent/descent speed of the drone varied between 1 and 2.8 m/s. Such vertical speed results in hysteresis loops (time lag) in the measured temperature vertical profiles when the response time of the sensors is not optimal (Cassano, 2014; Mašić et al., 2019). Slight differences in the temperature readings from the three RTD sensors can be explained by the different locations of the sensors on the drone airframe. The readings from the RTD attached under the left rear rotor (rotor 3 RTD) exhibit a systematic bias relative to the readings from the two other sensors (top and pole, see Figure 1 for locations of the RTDs on the drone frame). The bias is more visible during the drone's descent when in the presence of a steep temperature inversion the propulsion system pushes warmer air from above the drone downward and mixes it with colder air under the drone.

To correct for the time lag we followed the approach suggested by Cassano (2014) and introduced a fixed time shift between recorded altitudes and pole RTD temperatures to minimize the difference between the temperature profiles taken on the ascent and descent. For the measurements conducted on 28 February 2017 the optimal time lag was found to be 3.3 s which is 0.9 s smaller than the time constant of 1PT100KN1510 RTD element (4.2 s at 63.2% response at 1 m/s air flow speed according to the RTD element specification). Raw (black solid line) and corrected (red solid line) temperature profiles from the pole sensor are shown in Figure 5 (b).

It can be seen that according to the drone measurements a steep SBI is present in the first 10 m above the ground. The inversion becomes weaker above 10 m. To retrieve SBI lapse rates the corrected drone temperature profile was than averaged (blue solid line in Figure 5 (b)). The profile was split into two parts in terms of altitude above the ground (below and above 10 m) and a linear fit was applied to each part. The SBI lapse rates for 0-10 m and 10-50 m layers was found to be 32 °C/100 m (320 °C/km, blue dotted line in Figure 5 (b)) and 5 °C/100 m (50 °C/km, dashed blue line in Figure 5 (b)). Temperature profiles from the RS launched from the ECCC WS at 11:15 and 23:15 UTC and corrected for the altitude difference between the ECCC WS and the RTS takeoff/landing pad elevations are depicted in Figure 5 (b) for reference.

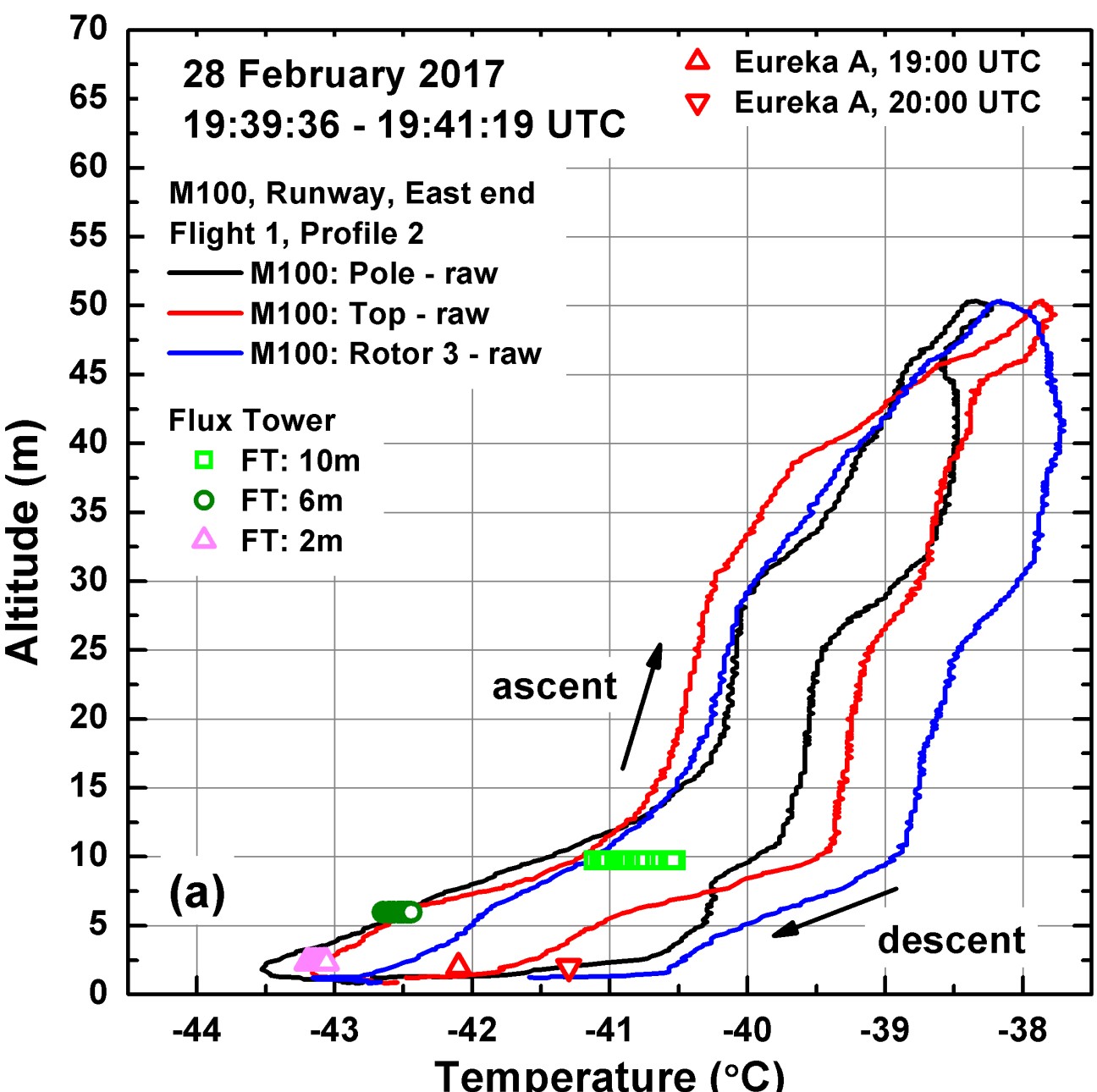

(a) a

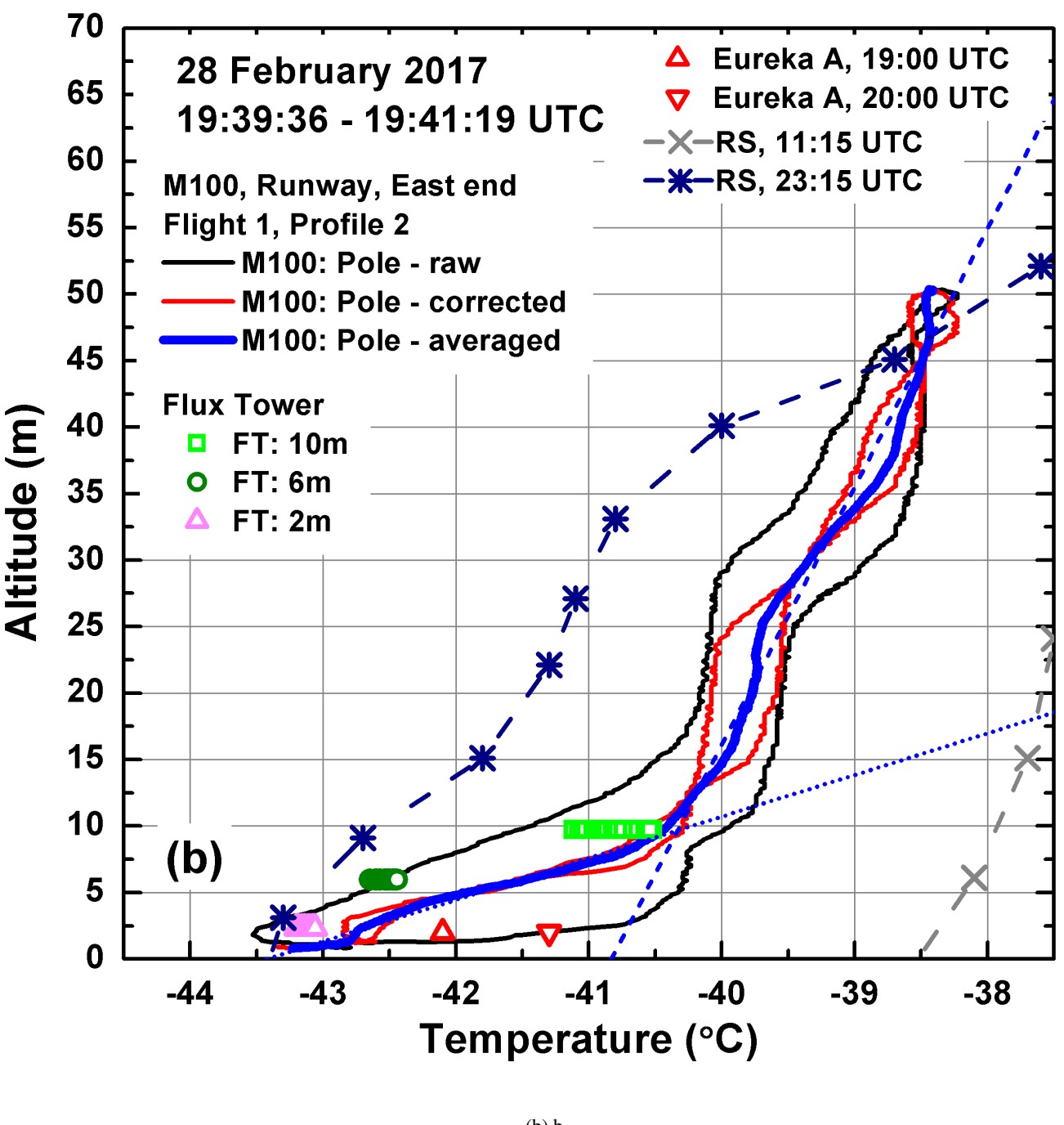

**Figure 5.** An example of raw (a) and time lag corrected (b) temperature profiles measured by M100 near the takeoff/landing pad 250 m south of the FT between 19:39 and 19:41 UTC on 28 February 2017. In subplot (b) blue dotted line represents 32 °C/100 m (320 °C/km) inversion lapse rate, blue dashed line represents 5 °C/100 m (50 °C/km) inversion lapse rate (see details in text).

Figure 6 shows Eureka A and 2, 6 and 10 m 3 Hz RTD FT temperatures (a), SBI lapse rates (b) retrieved from linear regressions of the FT temperatures as well as Eureka A and FT 1 minute wind speeds (c) between 19:30 and 20:30 UTC on 28 February 2017. The FT data taken during the time frame of three sets of M100 ascents and descents (19:37-19:43 UTC) are highlighted by thicker lines in Figure 6. Drone SBI lapse rates retrieved from three time lag corrected and averaged temperature profiles for 0-10 m layer (black symbols in Figure 6 (b)) are found to be in a good agreement with FT SBI lapse rates retrieved from the temperatures measured at 2, 6 and 10 m above the ground level (thick grey solid line in Figure 6 (b)).

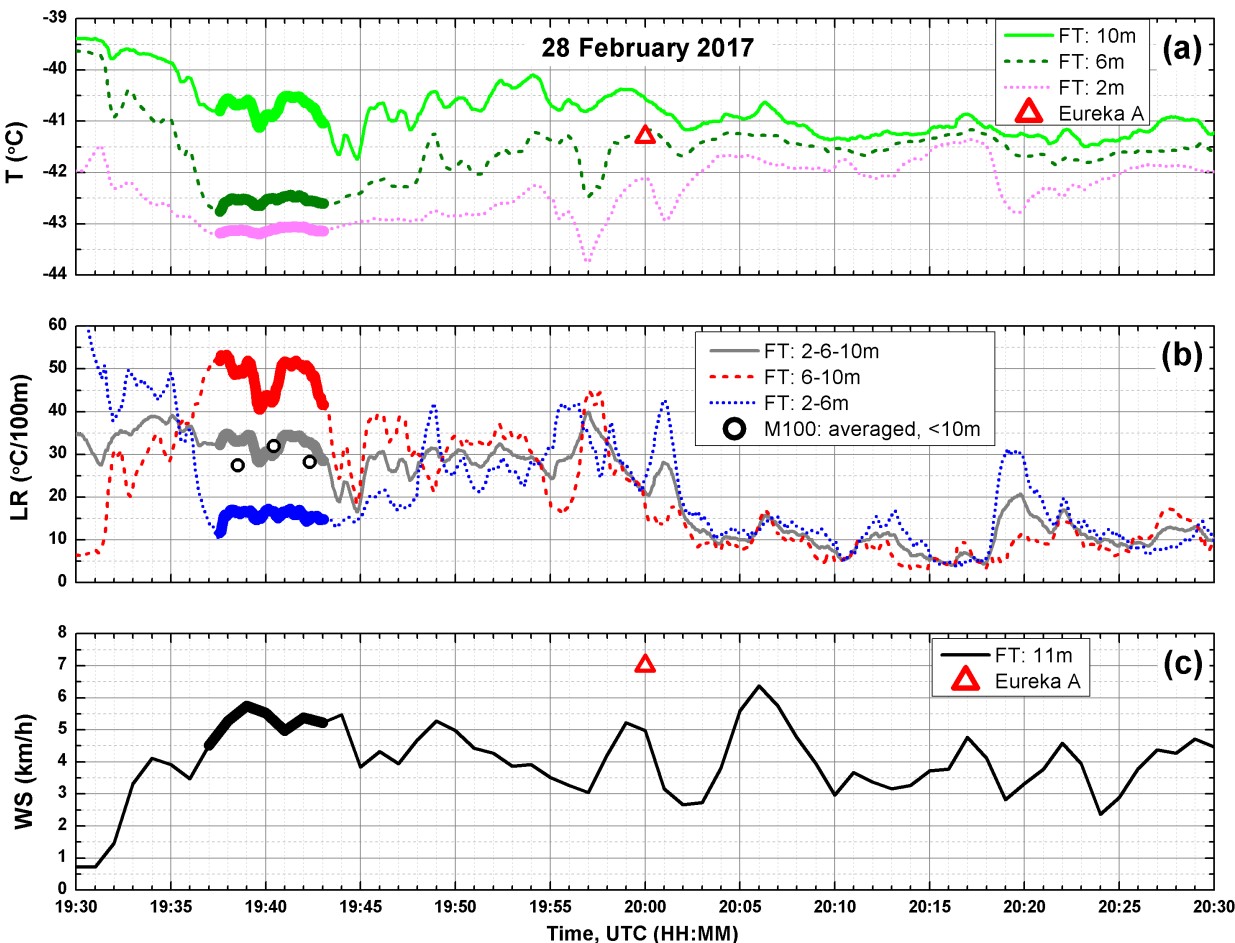

**Figure 6.** Time evolution of: (a) - air temperatures (T) from the FT 2, 6, and 10 m RTDs and Eureka A sensor, (b) - FT and M100 SBI lapse rates and (c) - wind speeds (WS) from 1 minute FT 11 m wind vane and Eureka A anemometer between 19:00 and 20:00 UTC on 28 February 2017. The FT data taken during the time frame of three sets of M100 ascents and descents (19:37-19:43 UTC) are highlighted by thicker lines.

The results of the tests conducted in February 2017 showed that the drone was able to provide reliable data at ambient temperatures below -40 °C. Drone temperature profiles and SBI lapse rates for 0-10 m altitude layer are in agreement with the data from the FT. Comparisons with RS data indicated some variations in the absolute temperatures and SBI lapse rates obtained from the instruments. While in Figure 5 the drone temperature profile has reasonably good match with the 23:15 UTC RS profile, a several degrees positive or negative bias can be observed between the profiles from day to day. First of all, this could be due to the time difference between the drone flights and RS launches, which were several hours apart. Secondly, the RS are launched from the ECCC WS. The RTS and the ECCC WS have ∼3.3 km horizontal separation from each other and are sitting at different elevations above the mean sea level. When the RS reaches the elevation of the RTS, it is ∼73 m above the launch site ground. Due to this and local topographic features, the SBI sensed by the RS could differ from the SBI sensed by the drone.

### 3.2 M210 RTK drone

#### 3.2.1 M210 RTK flight procedure

In 2018 the M100 was replaced by the M210 RTK. The main purpose of the replacement was to improve the positioning accuracy and enhance the stability during automatic flights. Before using M210 RTK in Eureka, the drone's flight performance was tested in Halifax. During the tests, the RPAS's navigation system managed to engage the RTK mode all the time and kept the positioning accuracy and stability of the drone within the specification. Also the tests showed that the drone provides equally good performance while flying either in RTK mode or in P-mode when RTK system is intentionally disabled.

Due to some technical problems with M210 RTK initial firmware and a few hardware failures, full scale operations in Eureka resumed only in 2020. Temperature measurements were conducted at both the RTS and the FTS. Typical flight time varied between 22 and 29 min per a set of two TB55 batteries. Since we had two sets of batteries available and it usually took ∼4 hours to recharge them with a standard charging hub (DJI IN2CH), M210 RTK temperature measurements were limited to two flights per day.

Also the original GNSS receiver (MediaTek, MT3339) of the data collection system used with M100 was replaced by a more advanced one built on u-blox MAX-M8Q concurrent GNSS engine (Zubax Robotics, 2019). It obtains position information from GPS, GLONASS and Galileo constellations simultaneously at up to 15 Hz sampling rate. Additionally, the pole temperature sensor was equipped with a small fan which provided continuous aspiration of the RTD element by forced air flow at ∼1 m/s speed to improve its response time.

To resolve hardware related biases of the temperature measurement system (RTD element production tolerance, MAX31865PMB1 module digitization errors), before the flights we conducted a laboratory test where all three RTDs were placed as close to each other as possible and aspirated with a room temperature air flow at a speed of ∼1 m/s using a fan. After the flights the RTDs were validated against the temperature measurements in the flowing water and melting ice. Pole temperature sensor was found to be the most accurate. Its absolute bias did not exceed -0.003±0.013 °C according to the results of the temperature measurements in the melting ice. Top and rotor 3 temperature sensor biases were found to be less than 0.25±0.02 °C and 0.30±0.02 °C,

correspondingly according to the results from the air flow, flowing water and melting ice tests. The biases were taken into account at the data post-processing phase to retrieve temperature values from our March 2020 drone measurements. In addition, the time lag was corrected following the same procedure applied to M100 data and described by Cassano (2014). Finally, high linearity and stability of platinum RTDs together with the results of validation of the sensors and application of bias correction

allowed us to conclude that the accuracy of our temperature measurements is ~0.3 °C. This value is equal to the required instrument measurement uncertainty recommended by WMO (Taalas, 2018) for below -40 °C air temperatures.

On 2 and 3 March 2020 four preliminary flights were performed in Eureka to test the drone and the flight procedure. The results of the preliminary flights showed that when the M210 RTK navigation system failed to engage RTK mode, which happened sporadically, the drone performance was somewhat similar to that observed with M100. Main symptoms of the

430 failure were circular shape flight tracks, fly away events and/or the inability of the drone to keep its altitude constant during the flight. Unfortunately, due to a "black box" type of the drone's navigation system it was not possible to find a solution. But when the RTK mode was engaged, the vertical and horizontal positioning accuracy of M210 RTK was maintained well within the manufacturer's specification.

Our measurement flights started on 5 March 2020. Between 5 and 9 March 2020 the flights were conducted at the RTS.

Each operation day consisted of two types of flight. The first type was an automatic flight with periodic ascents and descents along the preprogrammed way-points from the East end of the runway towards the FT (flux tower flight). An example of the way-points and flight trajectory is marked in black open circles and black solid line respectively in Figure 7. This type of flight was conducted to study the SBI and its temporal and spatial variability over a flat terrain (see subsection 3.2.2).

The second type was a manually controlled flight with a temperature profile measured in the gully close to the East end of

440 the runway following by a profile at the edge of the runway. An example of the flight trajectory is marked in blue solid line in Figure 7. This type of flight (gully versus runway flight) was conducted to investigate how local topography could influence the SBI (see subsection 3.2.3).

On 10 March 2020 at the end of the campaign two measurement flights were carried out on Slidre Fjord near the ECCC WS to study the SBI over the ice covered ocean (fjord flight, see subsection 3.2.4).

Table 1 summarizes the flights conducted between 2 and 10 March 2020 using the M210 RTK drone. All the measurement flights between 5 and 10 March 2020 were performed at low ascent/descent speeds (0.1-0.7 m/s) to further minimize the effect of the RTD response time on the temperature readings. The obstacle avoidance system of the drone was disabled during all flights. The 18:00, 19:00 and 20:00 UTC meteorological conditions are outlined in Table 2 for three locations: Eureka A, Eureka C and 0PAL. Unfortunately, due to a hardware failure no meteorological data were available from the FT for the time

period covering M210 RTK flights.

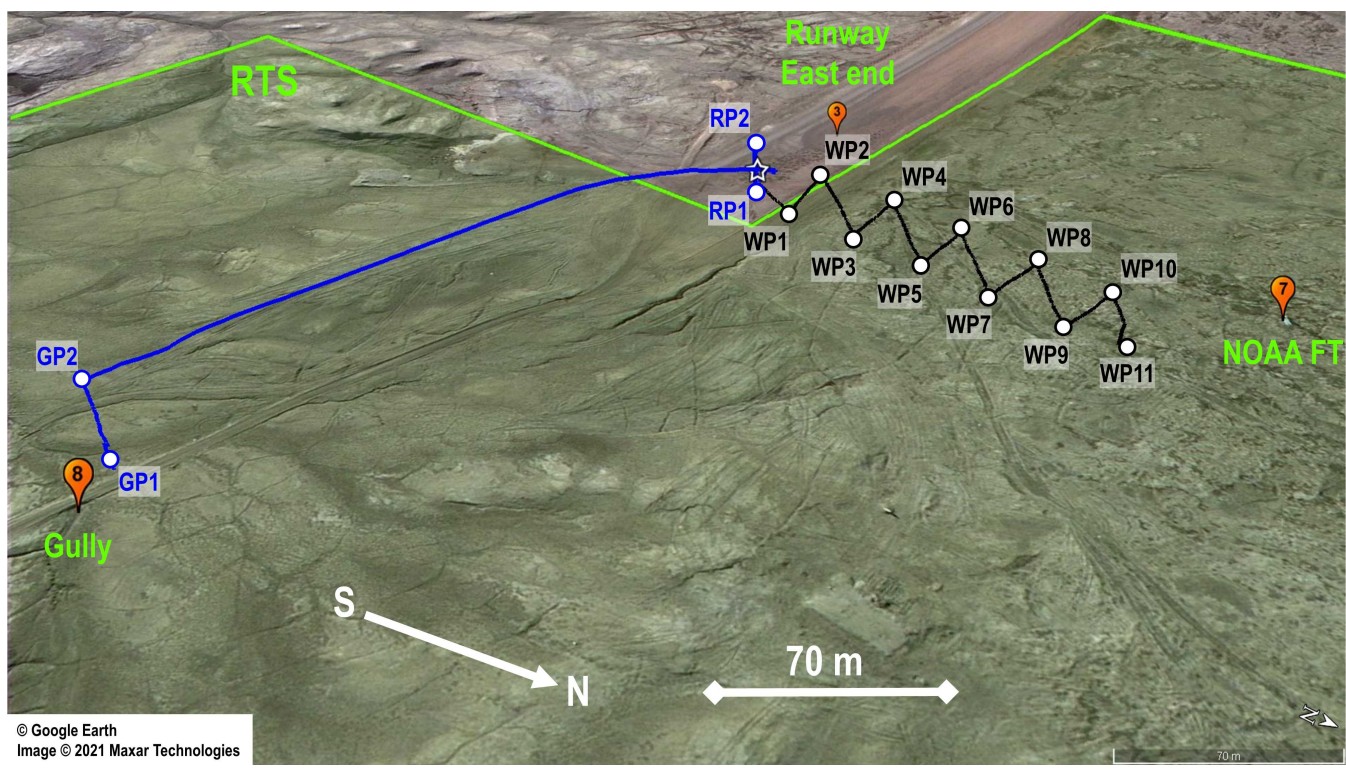

**Figure 7.** M210 RTK flux tower (black solid line) and gully versus runway (blue solid line) flight trajectories at the Runway Test Site (RTS) on 6 March 2020. Pins in the figure indicate the locations of the East side of the Eureka Aerodrome runway - #3, NOAA Flux Tower - #7, gully - #8. Typical way-points of the drone flights near the FT are marked in red open circles (WP1-WP11). Typical way-points of the gully versus runway drone flights are marked in blue open circles (RP1-RP2 - runway temperature profile, GP1-GP2 - gully temperature profile).

**Table 1.** M210 RTK flights in 2020.

| Date | Takeoff time (UTC) | Landing time (UTC) | Flight duration (min) | Type of operations | Average speed of ascent/descent (m/s) |
|---|---|---|---|---|---|
| 2 March 2020 | 18:22 | 18:51 | 29 | test flight near the FT | 2/3 |
| | 18:56 | 19:18 | 24 | test flight near the FT | 6/4 |
| 3 March 2020 | 18:29 | 19:55 | 26 | test flight in the gully | 3.5/- |
| | 19:01 | 19:23 | 22 | test flight near the FT | 0.2/0.2 |
| 5 March 2020 | 18:25 | 18:48 | 22 | 6 temperature profiles near the FT | 0.7/0.7 |
| | 18:52 | 19:15 | 23 | gully vs runway temperature profiles | 0.3/- |
| 6 March 2020 | 18:28 | 18:52 | 24 | gully vs runway temperature profiles | 0.1/- |
| | 18:52 | 19:15 | 23 | 5 temperature profiles near the FT | 0.3/0.3 |
| 7 March 2020 | 18:29 | 18:53 | 24 | gully vs runway temperatures profiles | 0.1/- |
| | 19:02 | 19:25 | 23 | 6 temperature profiles near the FT | 0.3/0.3 |
| 9 March 2020 | 18:30 | 18:54 | 24 | gully vs runway temperatures profiles | 0.1/- |
| | 18:59 | 19:23 | 24 | 3 temperature profiles near the FT | 0.4/0.3 |
| 10 March 2020 | 18:39 | 19:04 | 25 | 2 temperature profiles on the fjord | 0.2/0.2 |
| | 19:10 | 19:33 | 23 | 1 temperature profile on the fjord | 0.2/0.2 |

**Table 2.** Meteorological conditions at the Eureka Aerodrome (Eureka A), the Eureka Climate (Eureka C), and Zero Altitude PEARL Auxiliary Laboratory (0PAL) at 18:00|19:00|20:00 UTC during the measurement campaign in March 2020.

| Date | Location | Temperature (°C) | RH (%) | Wind direction (°) | Wind speed (km/h) | Visibility (km) | Pressure (hPa) | Conditions |
|------|----------|------------------|--------|--------------------|--------------------|-----------------|----------------|------------|
| 2 March 2020 | Eureka A | -29.8\|-30.3\|-30.1 | 74\|73\|73 | 5\|23\|12 | 5\|6\|7 | 24.1\|24.1\|24.1 | 992.9\|992.9\|993.0 | mainly clear |
|  | Eureka C | -31.8\|-30.8\|-30.2 | 75\|75\|76 | 12\|12\|9 | 8\|5\|8 | NA | 1003.1\|1003.1\|1003.2 | NA |
|  | 0PAL | -31.2\|-30.3\|-29.9 | 67\|67\|68 | 57\|79\|54 | 9\|5\|10 | NA | 1002.2\|1002.2\|1002.3 | NA |
| 3 March 2020 | Eureka A | -27.1\|-27.2\|-27.2 | 75\|75\|75 | 18\|8\|16 | 8\|6\|9 | 4.8\|8.8\|8.1 | 994.7\|994.9\|995.5 | snow |
|  | Eureka C | -27.6\|-26.1\|-27.3 | 79\|81\|79 | 10\|8\|9 | 7\|9\|7 | NA | 1004.9\|1005.1\|1005.7 | NA |
|  | 0PAL | -27.1\|-25.7\|-26.5 | 70\|70\|71 | 32\|359\|18 | 8\|9\|6 | NA | 1004.1\|1004.3\|1004.9 | NA |
| 5 March 2020 | Eureka A | -43.6\|-41.3\|-41.8 | 65\|66\|66 | 6\|36\|36 | 3\|1\|3 | 24.1\|24.1\|24.1 | 1012.7\|1012.6\|1012.8 | clear |
|  | Eureka C | -43.2\|-43.3\|-43.5 | 64\|64\|63 | 8\|13\|11 | 4\|3\|7 | NA | 1023.5\|1023.5\|1023.7 | NA |
|  | 0PAL | -44.3\|-44.2\|-43.4 | 64\|66\|65 | 40\|53\|39 | 9\|3\|7 | NA | 1022.6\|1022.6\|1022.8 | NA |
| 6 March 2020 | Eureka A | -43.2\|-43.9\|-43.1 | 65\|64\|64 | 6\|35\|36 | 5\|4\|2 | 16.1\|16.1\|16.1 | 1009.1\|1009.2\|1009.4 | ice crystals |
|  | Eureka C | -45.6\|-44.6\|-44.7 | 59\|60\|60 | 12\|8\|12 | 8\|3\|6 | NA | 1020.1\|1020.1\|1020.3 | NA |
|  | 0PAL | -45.1\|-43.8\|-44.5 | 64\|64\|63 | 54\|18\|62 | 8\|9\|6 | NA | 1019.1\|1019.2\|1019.3 | NA |
| 7 March 2020 | Eureka A | -45.8\|-44.7\|-44.3 | 63\|63\|63 | 4\|35\|36 | 4\|3\|1 | 24.1\|24.1\|24.1 | 1006.3\|1006.1\|1006.5 | clear |
|  | Eureka C | -46.1\|-46.0\|-45.9 | 60\|60\|60 | 10\|13\|9 | 9\|3\|3 | NA | 1017.1\|1017.1\|1017.2 | NA |
|  | 0PAL | -45.8\|-45.6\|-45.4 | 64\|61\|62 | 36\|39\|44 | 10\|5\|5 | NA | 1016.2\|1016.1\|1016.3 | NA |
| 9 March 2020 | Eureka A | -45.7\|-46.1\|-45.3 | 62\|62\|62 | 8\|9\|12 | 5\|4\|4 | 24.1\|24.1\|24.1 | 1002.9\|1002.9\|1003.2 | clear |
|  | Eureka C | -46.9\|-46.8\|-46.6 | 59\|60\|59 | 8\|12\|6 | 4\|5\|2 | NA | 1013.8\|1013.9\|1014.1 | NA |
|  | 0PAL | -47.6\|-46.8\|-46.9 | 62\|62\|63 | 2\|47\|41 | 12\|4\|6 | NA | 1012.9\|1013.0\|1013.2 | NA |
| 10 March 2020 | Eureka A | -44.3\|-42.4\|-43.6 | 63\|64\|63 | 36\|12\|1 | 1\|4\|4 | 24.1\|24.1\|24.1 | 1006.4\|1006.7\|1007.3 | clear |
|  | Eureka C | -45.6\|-46.0\|-45.6 | 61\|59\|60 | 12\|12\|10 | 5\|8\|2 | NA | 1017.4\|1017.7\|1018.0 | NA |
|  | 0PAL | -45.4\|-45.5\|-45.3 | 61\|61\|61 | 26\|62\|47 | 4\|6\|4 | NA | 1016.5\|1016.7\|1017.1 | NA |

### 3.2.2 *Flux tower flights: SBI variability*

Figures 8-11 show bias and time lag corrected temperature profiles measured on 5-9 March 2020 at various locations near the FT within the RTS. The measurements were conducted in clear sky conditions and at the wind speeds not exceeding 5 km/h for most of the time according to Eureka A meteorological station. Temperature profiles from the RS launched at 11:15 and 23:15 UTC from the ECCC WS together with 18:00, 19:00 and 20:00 UTC Eureka A temperatures are shown in Figures 8-11 for reference.

In the first flight on 5 March 2020 the drone was set to fly twice from the East end of the runway towards the FT along WP1-WP7 way-points and acquire three temperature profiles during each pass (see Figure 8, pass 1 and 2 along WP1-WP3 (a), WP3-WP5 (b) and WP5-WP7 (c)). For these measurements the optimal time lag was found to be 2.5 s. As it can be seen from the figure the temperatures dropped below -40 °C and a steep SBI was measured by the drone with an inversion lapse rate reaching ∼20-30 °C/100 m within the 0-10 m layer.

According to Figure 8, the bias and time lag corrected readings from the top and rotor 3 RTDs have 0.15 °C and 1.4 °C positive residue, respectively, in comparison with the readings from the pole RTD. The top RTD together with its MAX31865PMB1 module was plugged directly into the expansion board of the data collection system. It was located within a few centimetres from the Zubax GNSS and BMP280 modules. Internal temperature sensors of Zubax GNSS and BMP280 modules typically recorded temperatures which were 0.9 and 2.5 °C larger if compared to the pole RTD temperatures at -40 °C ambient temperature. We consider the heat produced by those modules and dissipated in the surrounding air could result in additional bias recorded by the top RTD. The rotor 3 RTD showed higher temperature during the flights, probably, because the heat generated by the spinning motor warmed up the air around it while the air was pushed downwards by the rotor 3 propeller and aspirated the RTD located below it. This result is in a good agreement with the findings reported by Greene et al. (2018), who studied the quality of the drone temperature measurements relative to the sensor locations on the airframe in a laboratory environment. The authors concluded that sensors installed right above or below the drone can be decoupled from the environment by stagnation in the air flow and can suffer from enhanced self-heating effects. Additionally, warm air streams caused by the spinning propellers can result in up to 1 °C positive bias in the temperature readings for the sensors located in proximity to the motors.

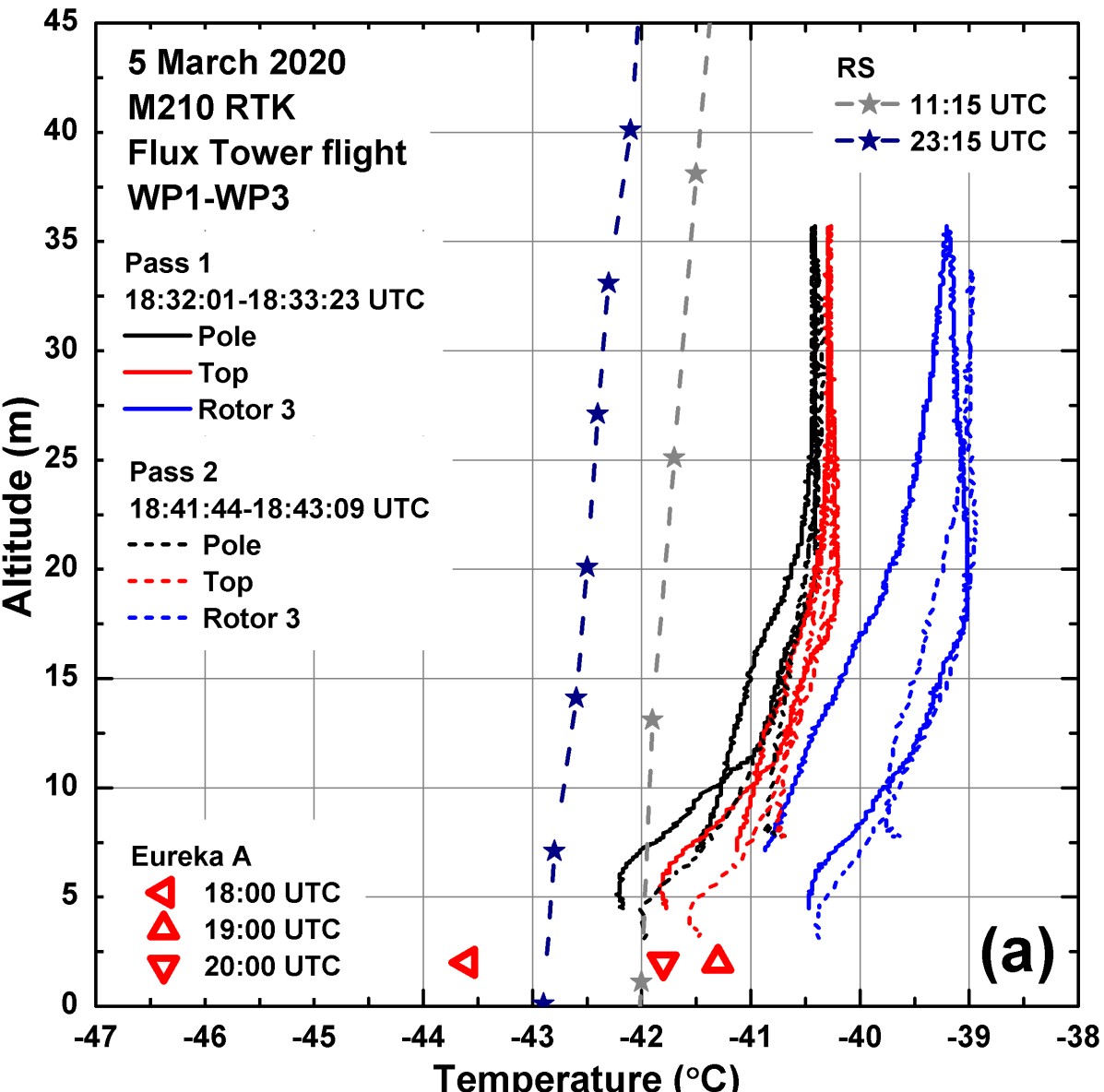

(a) a

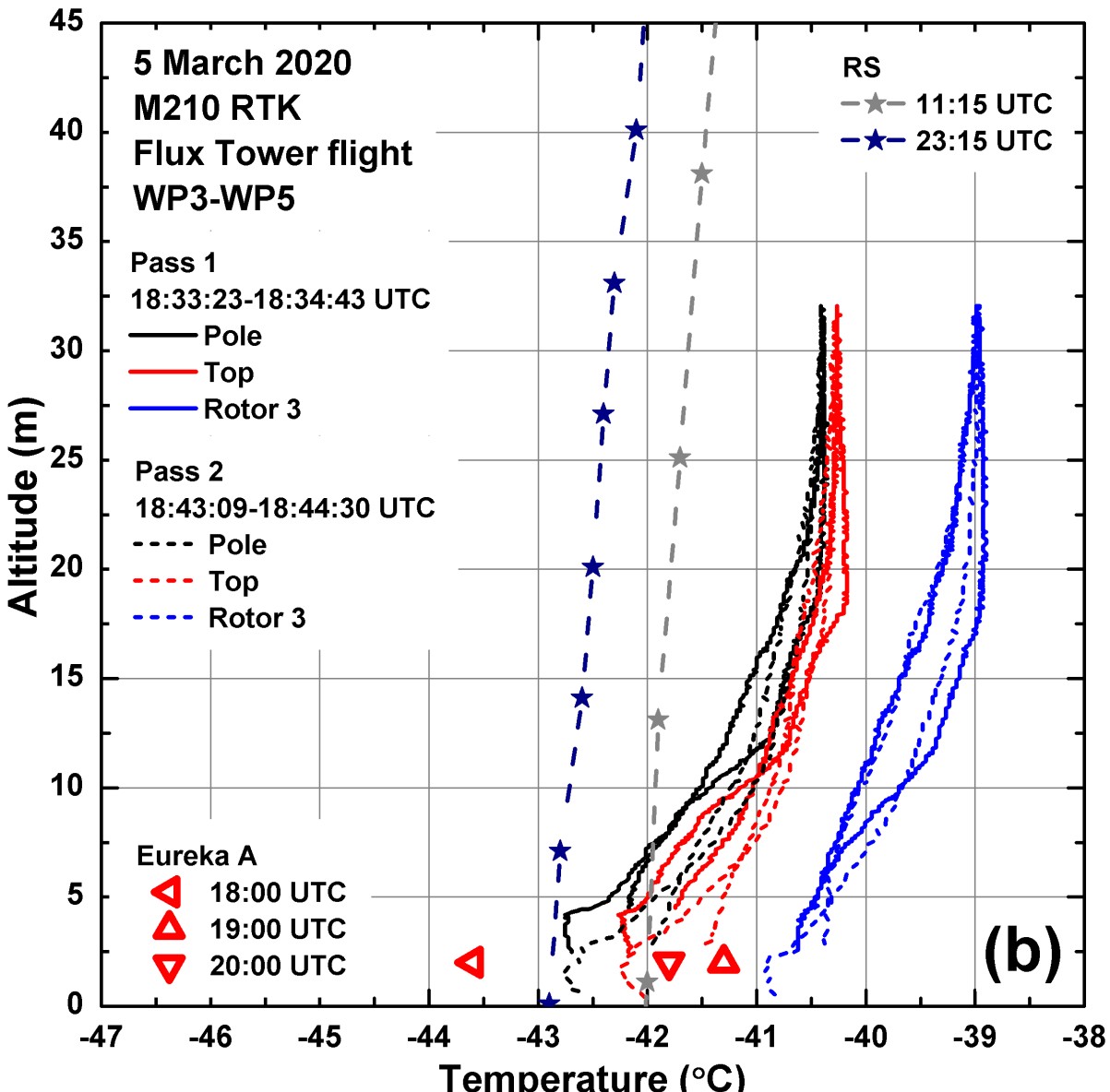

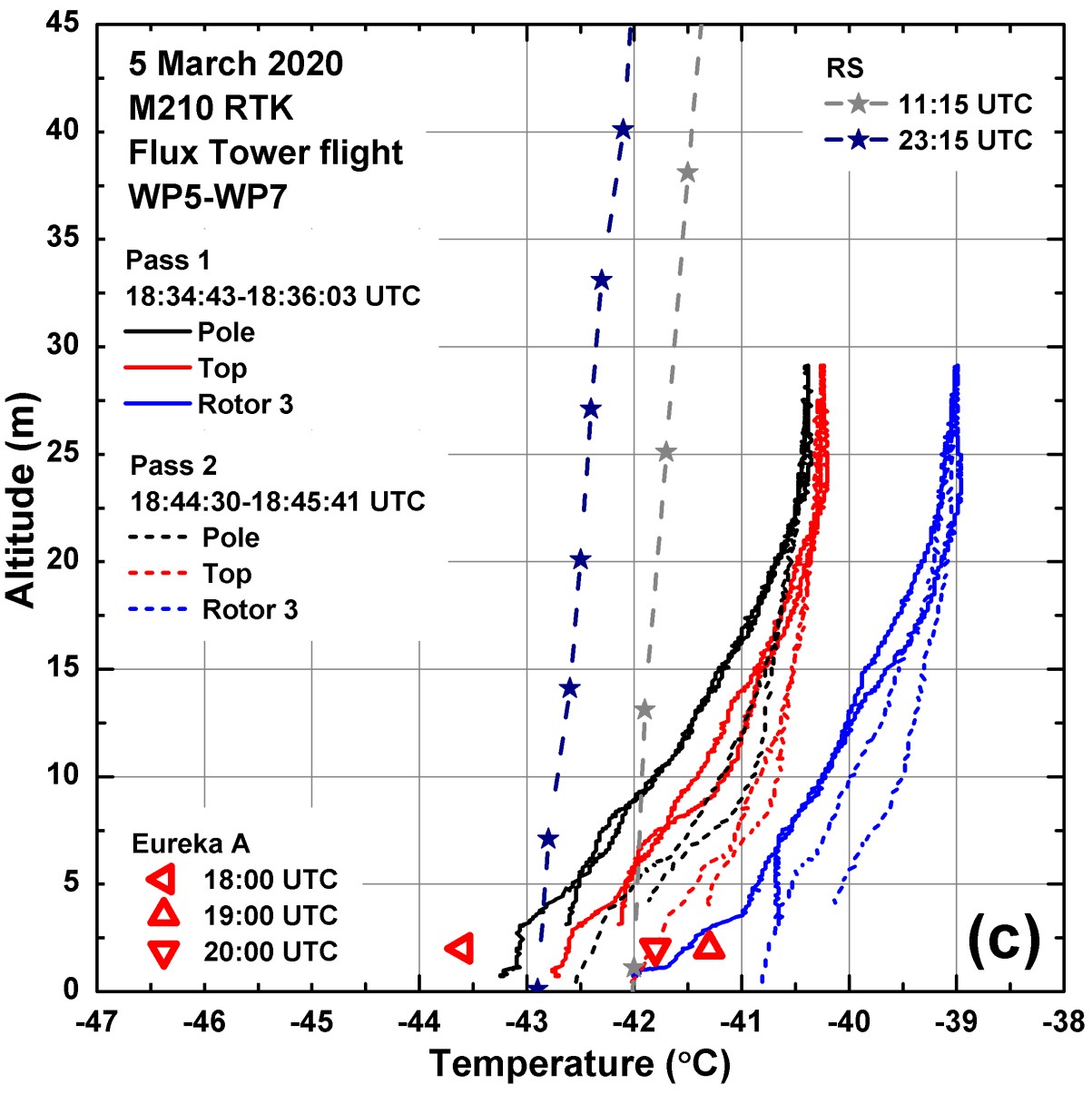

0

**Altitude (m)**

-47  -46  -45  -44  -43  -42  -41  -40  -39  -38

**Temperature (°C)**

**5 March 2020
M210 RTK
Flux Tower flight
WP5-WP7**

Pass 1
18:34:43-18:36:03 UTC
—— Pole
—— Top
—— Rotor 3

Pass 2
18:44:30-18:45:41 UTC
- - - Pole
- - - Top
- - - Rotor 3

Eureka A
◁  18:00 UTC
△  19:00 UTC
▽  20:00 UTC

RS
—★— 11:15 UTC
—★— 23:15 UTC

(c)

(c) c

**Figure 8.** Temperature profiles measured during M210 RTK two-pass flux tower flight along WP1-WP7 way-points (see Figure 7) on 5 March 2020 featuring biases between RTDs attached to different locations (color coded) on the air-frame and SBI temporal and spatial variability: (a) - WP1-WP3, (b) - WP3-WP5, (c) - WP5-WP7 (see Figure 7 for the locations of the temperature profiles).

Also, some difference is observed in the temperature profiles measured on the drone's ascents and descents and some artefacts are visible in the pole and top RTD profiles around 5 m altitudes and below. In the isothermal region above 20 m the ascent/descent temperature differences are less noticeable for the top and pole sensors in comparison with the rotor 3 sensor. This could be due to a combination of factors which include errors introduced by the fixed time lag correction, drone vertical and horizontal speed fluctuations, air mixing produced by the drone propellers as well as natural temperature variations. The differences in the rotor 3 RTD readings become more noticeable during the drone's descent into the steeper SBI (Figure 6 (a) and (b)). The propulsion system pushes warmer air from above the drone downwards and mixing it with colder air under the drone (downwash flow). This effect is similar to one observed during the M100 test flights in 2017. Additionally, some slight variation in the readings is visible in the profiles measured on the first and second pass conducted 9 minutes apart from each other. This is attributed to a change in the ambient conditions over time, since according to the FT temperature data (see Figure 6) natural temperature fluctuations of ∼1 °C per minute occur nearly continuously during periods of extremely stable boundary conditions in Eureka. Lampert et al. (2020a) also noticed up to ∼2 °C differences in the time lag corrected temperature profiles taken on the ascents and descents during the studies conducted near Greenland in September-October 2017 using their ALICE quad-copter. They attributed the differences to small variations in the meteorological conditions at higher altitudes (70-1000 m) occurred within few minute time intervals as well as to the heat generated by a local heat source (research vessel) and dissipated in the air at lower altitudes (<70 m). The temperature sensors installed onboard ALICE drone were TSYS01 and a fine wire RTD.

To make the subsequent figures easier to read, only the measurements from the pole RTD are presented further in the paper. After the temperature sensor validation tests in melting ice, this RTD was considered to be the most accurate sensor onboard our drone for air temperature measurements. It was forcibly aspirated and located away from the drone heat sources (motors, batteries and large electronics).

On 6 and 7 March 2020 two more flights were performed (Figures 9 and 10). The profiles show similar SBI pattern for both cases. On 6 March 2020 the SBI was steeper below 7-10 m with the inversion lapse rate reaching 20 °C/100 m in comparison to the SBI on 7 March 2020, which featured larger variability between individual temperature profiles. Above 10 m more gradual and close to isothermal temperature dependence is observed in both instances. For 7 March 2020 the average temperature was by about 1 °C lower along the entire altitude range in comparison with 6 March 2020. A ∼1.5 °C temperature drop was also registered by the Eureka A station and RS. On each day between 5 and 7 March 2020 a 2-2.7 °C positive bias in the temperatures measured by the drone at 30 m altitude above the ground relatively to those measured by the RS launched at 23:15 is observed. Both the drone and the RS recorded a similar inversion lapse rates (0-2 °C/100 m) in the altitude range higher than 30 m above the ground.

On 9 March 2020 the drone temperature profiles did not show as steep a SBI below 10 m even though the temperatures were lower and remained below -42 °C along the altitude range of the drone measurements (Figure 11). However, according to the RS measurements, the temperature did not change significantly in comparison with 7 March 2020 and the profiles obtained with the drone and the RS on 9 March 2020 are found to be in a good agreement featuring the inversion lapse rate of ∼3 °C/100 m within 5-55 m layer. During this flight the RTK system experienced a number of intermittent failures and the

drone was not able to maintain its altitude properly. Due to that the flight had to be restarted several times. The drone managed to complete the measurements of only three temperature profiles and performed only one ascent/decent which covered below 10 m altitude range before the battery drained. For 6 and 7 March 2020 FT flights the optimal time lag was found to be 2 s, while for 9 March 2020 FT flight such time lag correction resulted in increased difference between the ascent/descent temperature profiles and was neglected.

Most of the drone temperature profiles measured between 5 and 9 March 2020 show positive bias in comparison with the RS profiles (see Figures 8-11). For 5 March 2020 the drone pole temperatures at ~2 m level above the ground were found to be ~0.5 °C warmer than 18:00 UTC Eureka A temperatures. Eureka A temperature variations over 3 hours between 18:00 and 20:00 UTC totalled 2.3 °C. For 6-9 March 2020 close to the ground drone pole temperatures were up to ~2 °C warmer than 18:00, 19:00 and 20:00 UTC Eureka A temperatures. Again, we attribute these differences to the horizontal and vertical

separation between the measurement sites, time difference between the measurements and natural temperature variations.

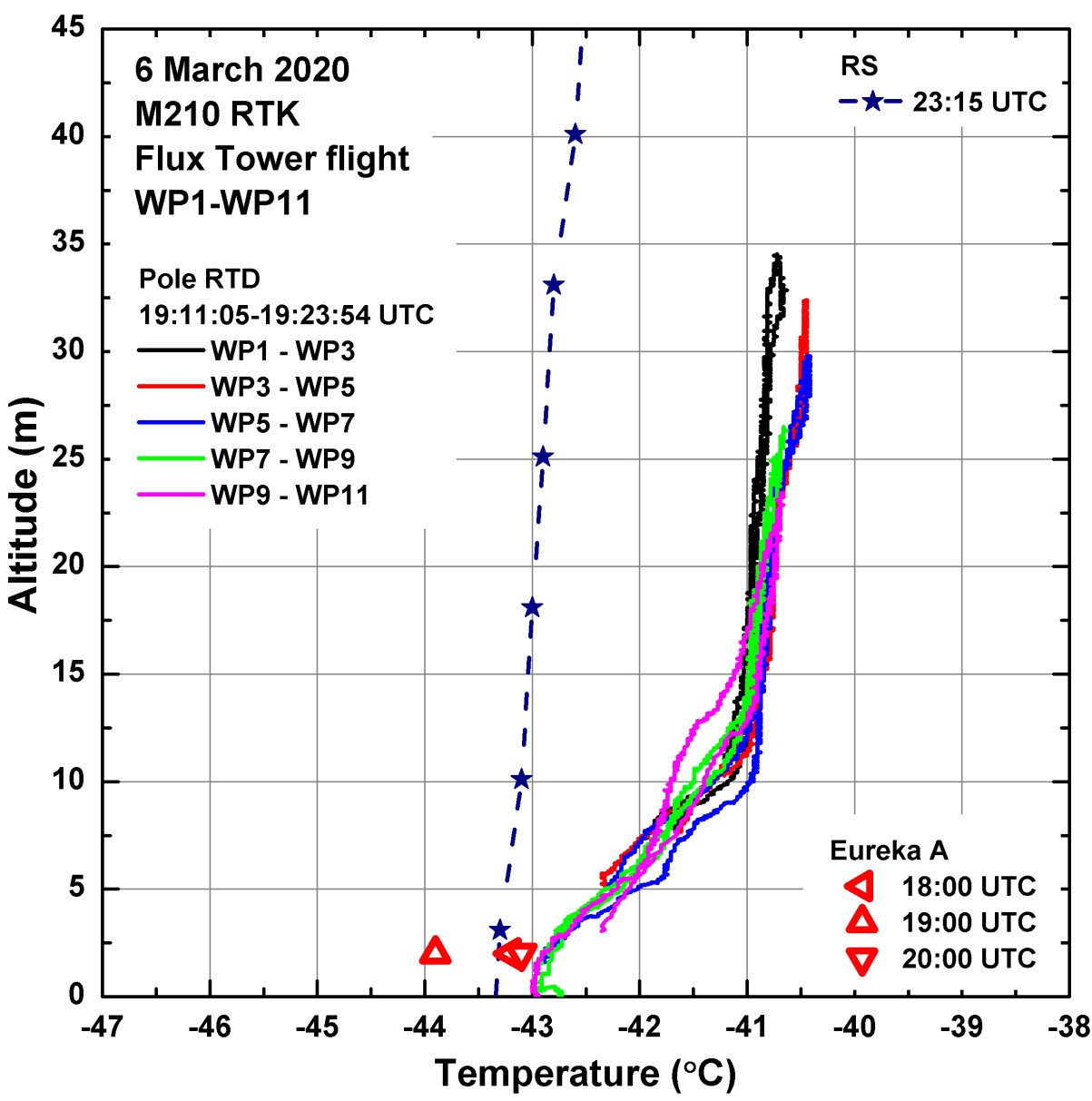

**Figure 9.** Temperature profiles measured during M210 RTK flux tower flight along WP1-WP11 way-points on 6 March 2020. Line colors represent individual profiles in the course of the flight (see Figure 7 for the locations of the temperature profiles).

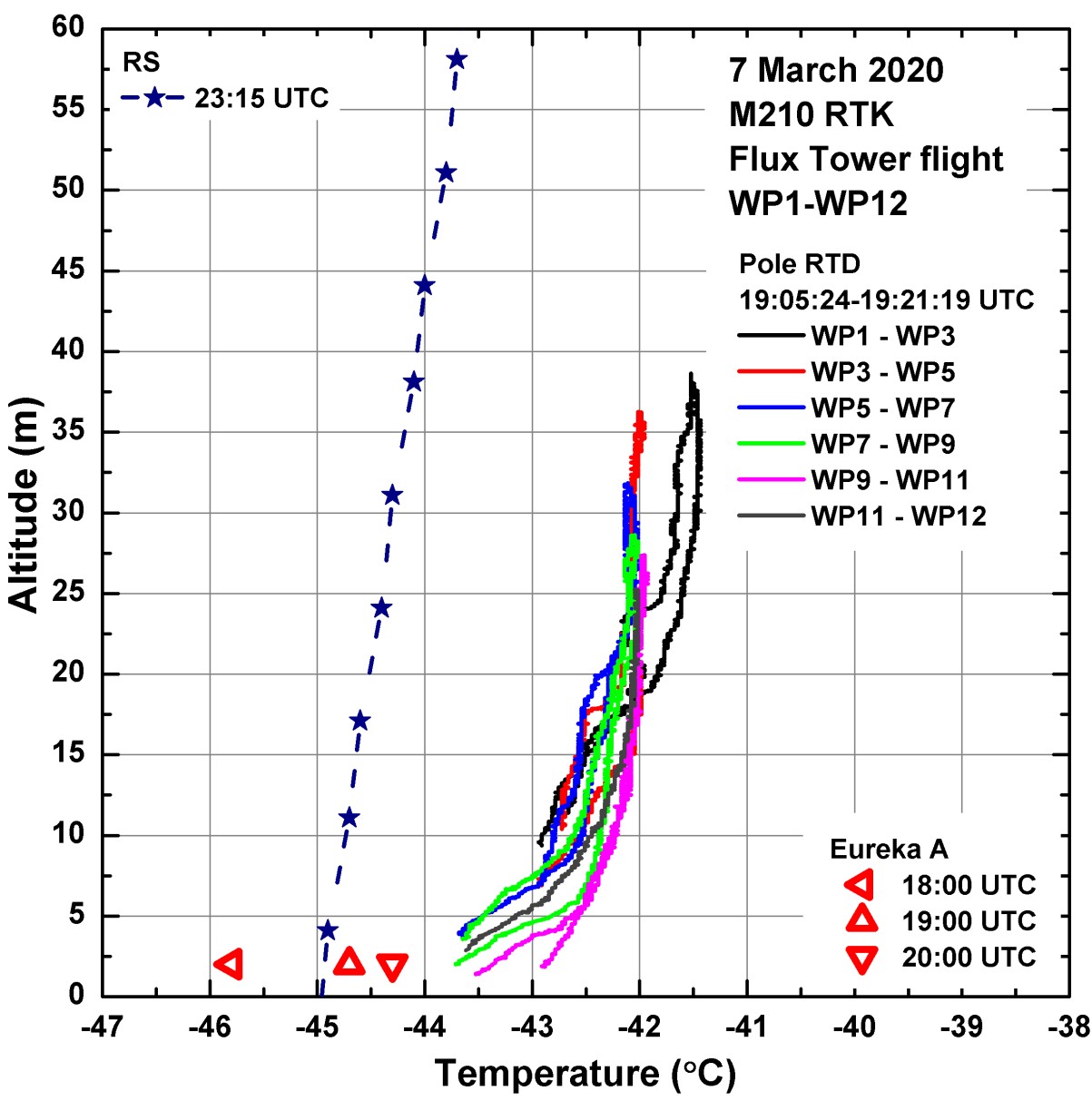

**Figure 10.** Temperature profiles measured during M210 RTK flux tower flight along WP1-WP12 way-points on 7 March 2020. Line colors represent individual profiles in the course of the flight (see Figure 7 for the locations of the temperature profiles).

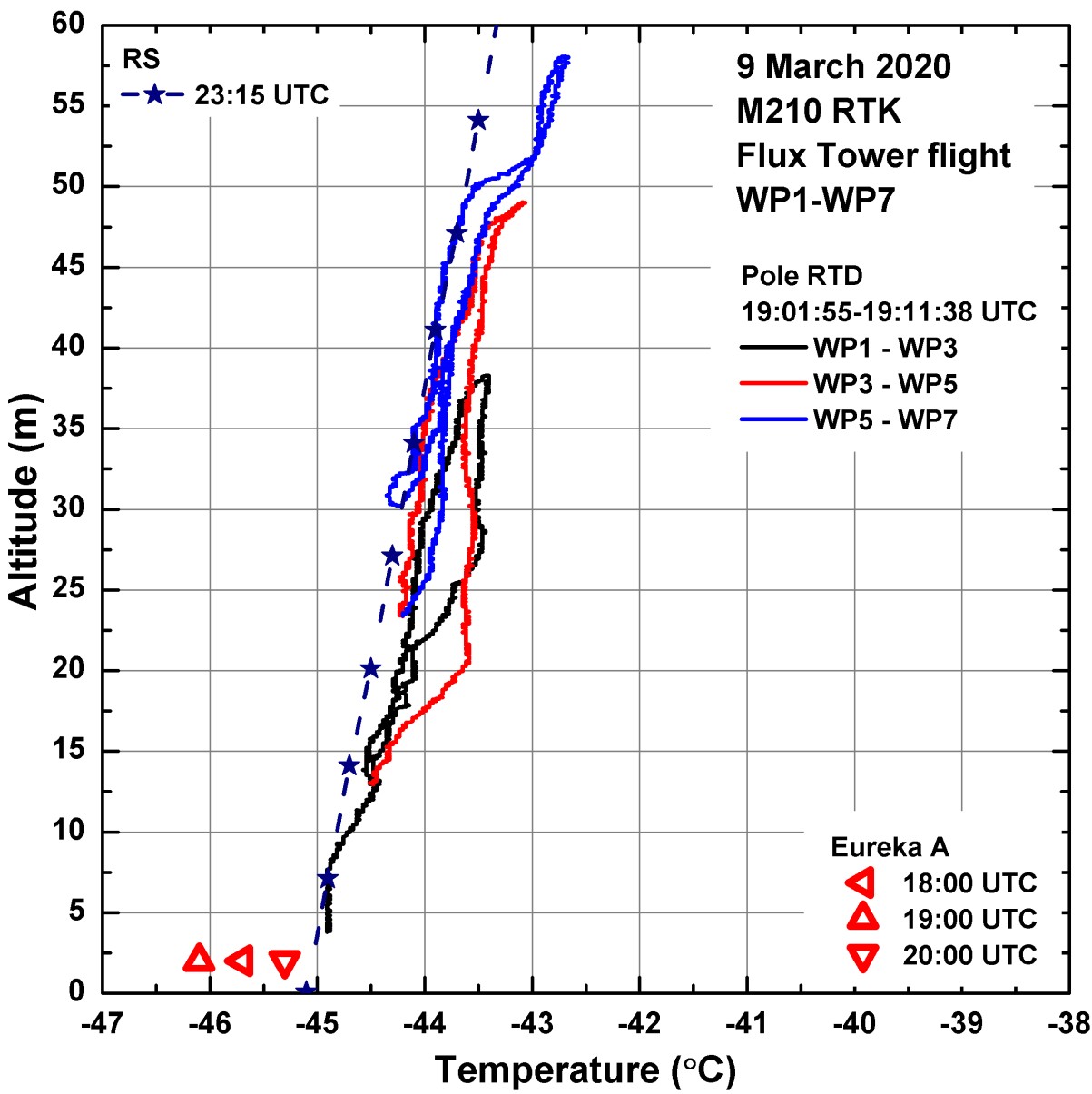

**Figure 11.** Temperature profiles measured during M210 RTK flux tower flight along WP1-WP7 way-points on 9 March 2020. Line colors represent individual profiles in the course of the flight (see Figure 7 for the locations of the temperature profiles).

### 3.2.3 *Gully versus runway flights: SBI and local topography*

Figures 12-13 demonstrate runway and gully temperature profiles measured on 5-9 March 2020. The gully's lowest point was located ∼30 m below the surface level of the runway. Preliminary flights showed that due to large altitude span and limited flight time it is not possible to complete ascent/descent profiles in both the gully and above the runway at low vertical speed on one set of batteries. Also to keep better track of the remaining capacity of the batteries and to maintain vertical speed at constant value it was more convenient to conduct the measurements on the ascents starting from the ground, rather than on the descents starting from some altitude level. Because of that, the temperatures shown in Figures 12-13 are those obtained on the drone's ascents only. Average ascent speed was kept at 0.1-0.3 m/s. An application of 2 s time lag correction did not result in any improvements in the profiles and was omitted for these flights.

On 5 and 9 March 2020 (Figure 12) the profiles measured at the runway smoothly extends the profiles measured in the gully. For both days the runway and gully profiles are close to each other in the 0-50 m region. However, for 5 March 2020 the SBI was steeper in the gully (∼10 °C/100 m inversion lapse rate), but more gradual above the runway. For 9 March 2020 the shapes of the SBI at the runway and in the gully are similar to each other, while it was generally by 2 °C colder at the gully surface in comparison with the runway surface suggesting that colder air pools in the gully depressions.

On 6 and 7 March 2020 (Figure 13) a different SBI regime was observed. The profiles are close to each other in shape but shifted vertically by an amount equal to the gully depth. This suggest that the local radiative cooling was responsible for both profiles and that there was no air flow interaction between the gully and runway sites. In contrast on 5 and 9 March 2020 (Figure 12) it appears that a weak surface horizontal air flow advected from the runway to the gully created similar temperature profiles.

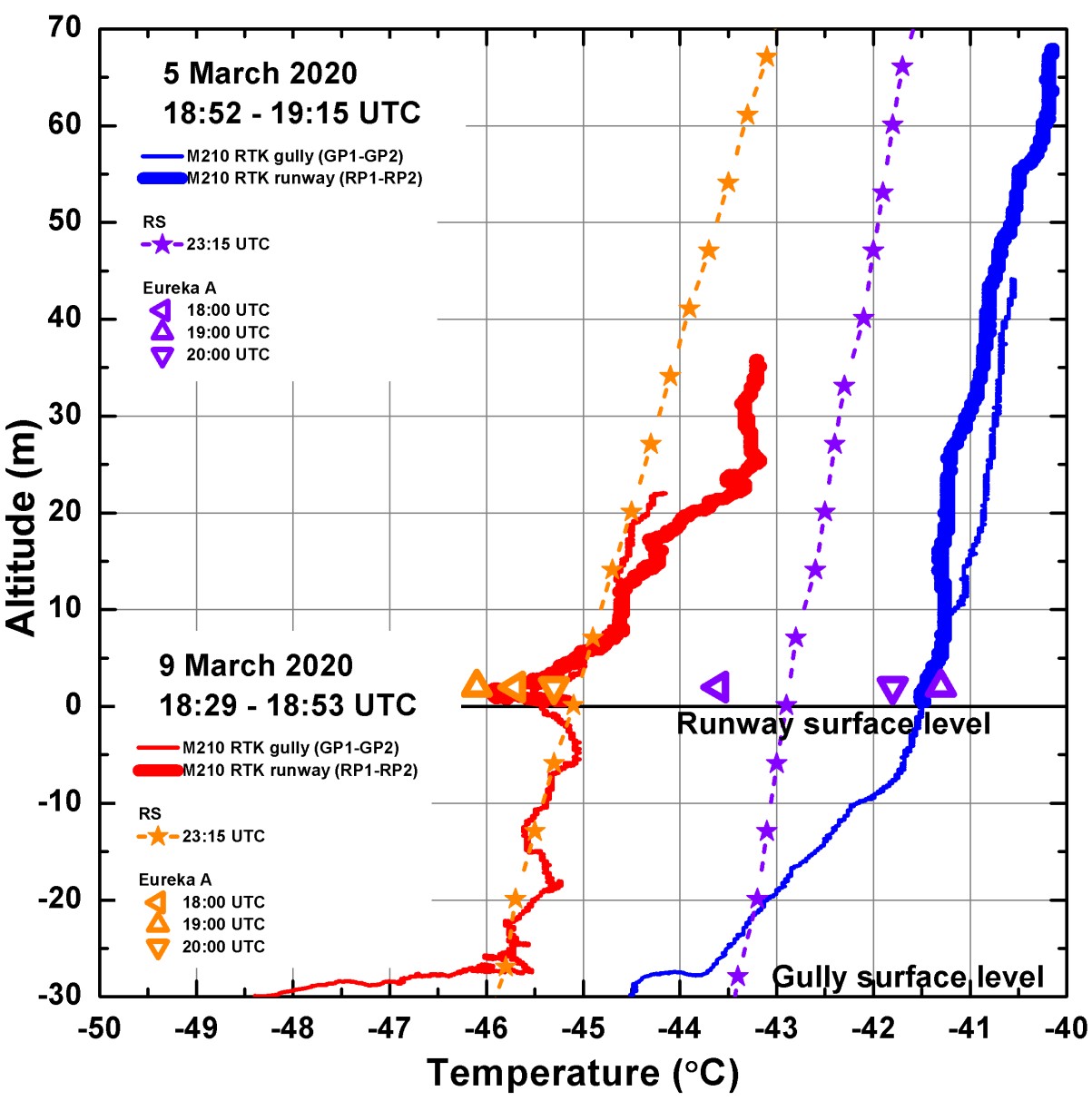

**Figure 12.** Temperature profiles measured during M210 RTK gully versus runway flights on 5 and 9 March 2020 featuring different shapes of SBI formed in the gully and above the runway (see Figure 7 for the locations of the temperature profiles).

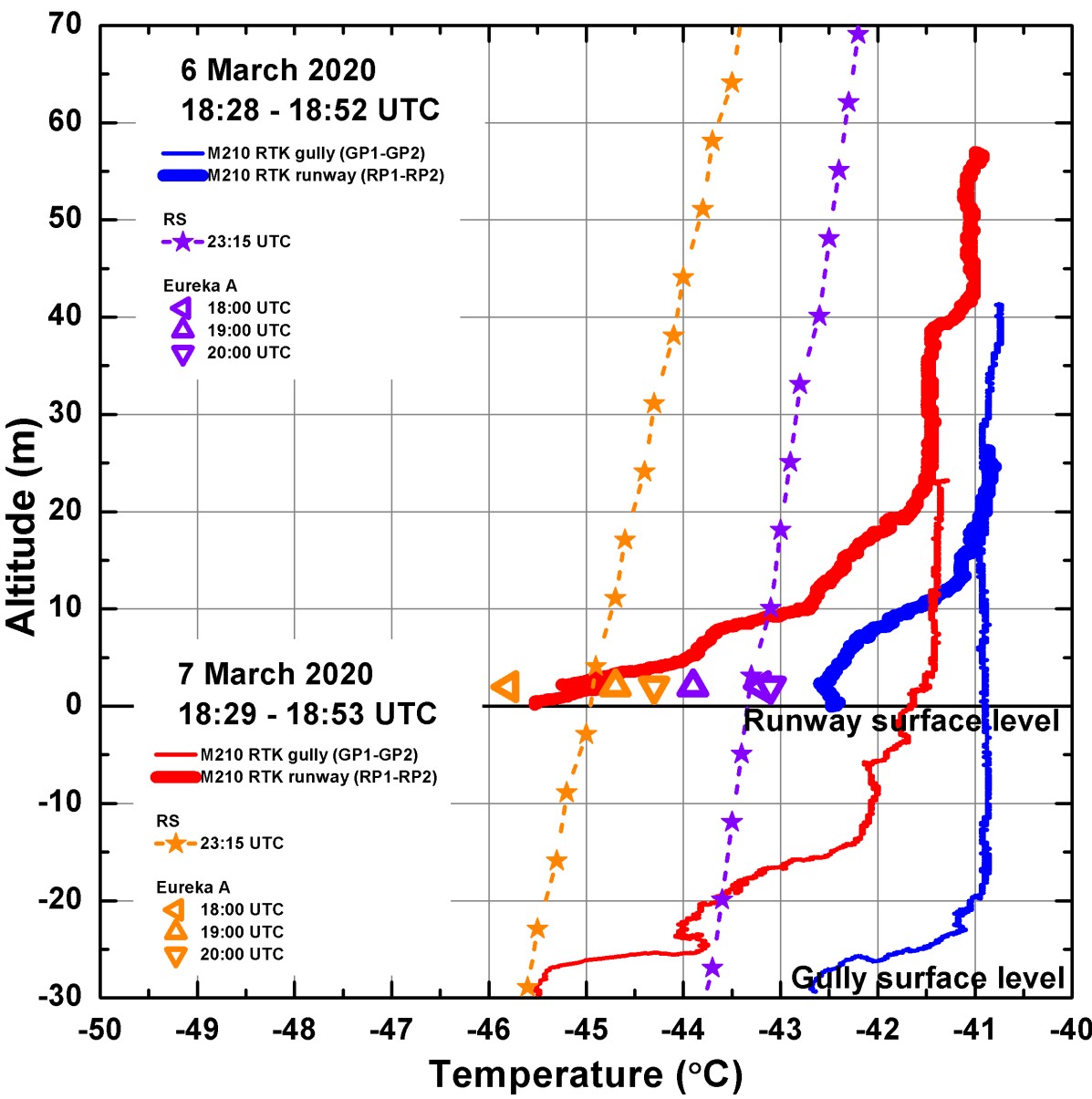

**Figure 13.** Temperature profiles measured during M210 RTK gully versus runway flights on 6 and 7 March 2020 featuring similar shapes of SBI formed in the gully and above the runway (see Figure 7 for the locations of the temperature profiles).

### 3.2.4 *Fjord flights: detection of ocean heat flux through the ice*

Figure 14 shows temperature profiles measured on 10 March 2020 on Slidre Fjord at various distances from the shoreline. The measurements were carried out to investigate the features of the temperature profiles above the ice covered ocean caused by heat flux through the sea ice (Pavelsky et al., 2011). According to the ice survey conducted on 6 March 2020 by the staff at the ECCC WS, the ice thickness was $\sim$1.9 m (Ice Thickness Program, Canadian Ice Service). All three profiles measured at $\sim$35, 210 and 414 m from the shoreline feature an isothermal atmospheric layer between 10 and 30-40 m. Above 40 m the SBI is characterized by $\sim$5 °C/100 m inversion lapse rate, while in the region between 0 to 10 m above the ice surface an unstable layer can be seen. In this case drone measurements were conducted within 500 m from the RS launch site and Eureka C meteorological station. Elevation separation between the sea ice level and ground level of the RS launch site and Eureka C was $\sim$10 m. Drone temperature profile measured $\sim$210 m from the shoreline (pin #6 in Figure 3) is found to be in a good agreement with 23:15 UTC RS profile in the altitude range between 30 and 55 m above the sea ice. Temperatures measured by the drone at 12 m above the sea ice $\sim$210 and $\sim$414 from the shoreline (pins #6 and #7 in Figure 3) agree within $\pm$0.5 °C with the temperatures measured at Eureka C site at 18:00, 19:00 and 20:00 UTC.

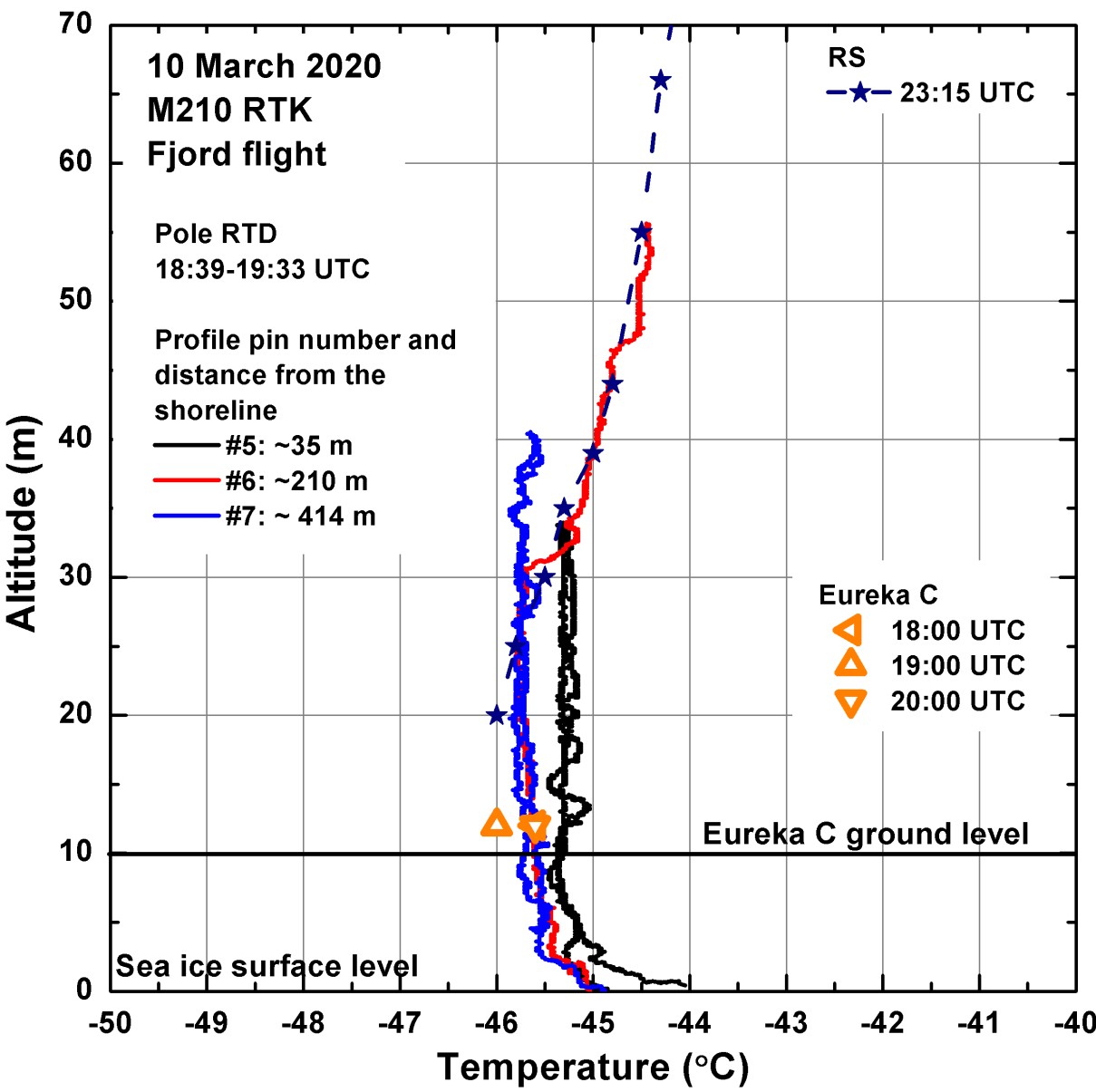

**Figure 14.** Temperature profiles measured during M210 RTK fjord flight on 10 March 2020 featuring the SBI over the ice covered ocean (see Figure 3 for the locations of the temperature profiles).

### 3.2.5 *Lessons learned and future prospects*

Although our sensors and data collection system allowed measurements at low temperatures, the response time of our RTDs did affect the temperature readings. The use of fine wire type RTDs would be beneficial since they have faster response (Wildmann et al., 2013), however their readings could suffer from temperature fluctuations due to turbulent flows caused by spinning propellers and would require additional filtering of raw data (Greene et al., 2018; Lampert et al., 2020a). Furthermore, we found that special care has to be taken to the temperature sensor mounting since the mechanical vibrations induced in the drone's air-frame during the flight tends to break the leads of our RTD elements.

Conducting the flight operations while residing inside a truck parked near by the flight region worked well for the drone pilot in control. We will continue this practice further during our winter field operations in Eureka.

Our results show the drone's flight time per a single battery charge was within the value specified by the manufacturer. However, as we would expect, the number of available batteries, their power capacity and time required for recharging became the key factors limiting our airborne time, hence, the amount of temperature measurements conducted per day. This can be solved by establishing a larger bank of spare batteries and utilization of multiple or more advanced chargers.

In the future, our payload can be improved by an installation of a non-contact infrared thermometer looking downwards, a laser altimeter and temperature sensors with better response time. The thermometer would allow measurements of the skin temperature simultaneously with the SAT, while the laser altimeter would provide data on the drone's altitude above the ground with a precision better than barometric altimeter. The altimeter also can be used to track fine scale topography of the surface during the flight. Forcibly aspirated fast temperature sensors placed away from the drone heat sources and areas of stagnated air flow are a key tool in providing reliable readings. Additionally, a development of robust time lag correction method, similar to one applied for the RS temperature (Mahesh et al., 1997) and humidity (Miloshevich et al., 2004) measurements, but more specific to the sensors installed onboard multi-rotor RPAS is required. To derive true air temperatures from the sensor readings and to obtain accurate scientifically useful results from the drone measurements the method should account for the variable ventilation rate of the sensor and, hence, for the variable time lag (Lampert et al., 2020a). To measure air temperatures within 0-10 m above the ground using multi-rotor RPAS an attention has to be paid to the development of an optimal flight strategy, which would minimize the effects of downwash, produced by the drone propellers. In these terms, horizontal flights at the lowest possible altitudes, allowing to sense undisturbed air close to the ground, seem to be more preferable than ascent/descent type flights used in our study. All these aspects have to be carefully considered during future mission planning and equipment integration stages.

## 4 Conclusions

We have reported on the application of two commercial drones made by DJI to investigate the SBI within 75 m of the ground in the harsh environment of High Arctic winter at ambient temperatures down to -46 °C. The results of test flights conducted with the M100 drone revealed issues in its navigation system, which made automatic flights in Eureka almost impossible. The issues were related to frequent losses of GNSS lock as well as poor performance of compass and barometric altimeter in high

latitudes and at low temperatures. This resulted in many occasions when M100 failed to maintain its altitude above the ground, drone fly away events and circular shape flight tracks when flown automatically along the preprogrammed set of way-points in P-mode. The M210 RTK drone equipped with RTK navigation system performed better than M100 and allowed us to conduct automatic flights in RTK mode.

For the flux tower flights, drone altitudes did not exceed a maximum of 60 m above the ground. For the gully versus runway flights, they did not exceed 75 m and 70 m above the gully and runway surface levels correspondingly. For the fjord flights, drone altitudes did not exceed a maximum of 55 m above the ice surface.

There are three reasons such altitudes were used:

  − In our SBI measurements we focused our interest on 0-100 m layer where strong temperature gradients are observed.
− Since the flights were conducted at low ascent/descent speeds (0.1-0.7 m/s), our priority was to measure as many temperature profiles per flight as possible before the drone batteries were drained. Due to this, in the majority of the flux tower flights, drone altitudes did not exceed a maximum of 40 m above the ground.
  − To comply with the Canadian Aviation Regulations, during our 2017 and 2020 flight campaigns drone maximum altitudes were kept within 91 m (300 ft) and 122 m (400 ft) above the ground level, correspondingly.

Our results show that multi-rotor drones can be effectively used in the High Arctic to characterize SBI and its temporal and spatial features. Our drone temperature profiles are in agreement with the temperatures from the FT measured at 2, 6, and 10 m above the ground and Eureka A temperature. The inversion lapse rates within the 0-10 m layer can reach the values of ∼10-30 °C/100 m (∼100-300 °C/km). This is about half the lapse rate measured by Hudson and Brandt (2005) for 0-2 m altitude layer above the snow surface on the Antarctic Plateau. In the 10-75 m layer above the ground the SBI is characterized
by smaller inversion lapse rates, which are in the range of ∼2-4 °C/100 m (∼20-40 °C/km) or less. In this region our drone lapse rates agree well with the lapse rates obtained from the Eureka RSs, launched within 4 hours after the drone flights. Also our 10-75 m drone lapse rates are close to the results of multi-year studies conducted by Bradley et al. (1993) and by Walden et al. (1996) based on Eureka RS data covering 1967-1990.

Comparisons of the results of the SBI measurements conducted in the gully and above flat area near the East end of the
runway suggest that local topography and a change in the micro-meteorological conditions could be factors shaping the inversion in the gully. Above the sea ice, the temperature profiles are found to be isothermal above a shallow unstable surface layer revealing the impact of the heat flux through the ice. A detailed study with thorough analysis of the FT and drone temperature data as well as heat flux data are required for better understanding of processes responsible for the inversion formation above the ground and sea ice surface.

For the flux tower and fjord flights our optimal temperature sensor time lag was found to be 3.3 s and 2-2.25 s for naturally aspirated (2017 flights) and forcibly aspirated (2020 flights) pole RTD, correspondingly. Our time lag values are close to 2-5 s time lag reported by (Cassano, 2014) for their naturally aspirated Pt 1000 Heraeus M222 and Sensirion SHT 75 sensors installed at SUMO fixed-wing drone. However, the measurement conditions, which include different sensor type and much larger RPAS air speeds used during their studies, make direct comparisons challenging (SUMO drone cruise air speed is 15 m/s).

Our results confirmed the findings reported by Greene et al. (2018) and showed that when a sensor is installed onboard a drone for the air temperature measurements, the most critical factors affecting the accuracy and responsivity of the sensor are its time constant and location.

Drone field studies of SBIs have the advantage of providing a rapid three-dimensional picture of the air temperature distribution. This allows one to identify spatial and temporal changes in the inversion lapse rates with altitude that cannot be captured by a fixed position flux tower. Also, the ability of the multi-rotor drones to perform flights and measure temperatures in the near surface layer, provides a high potential for micro-meteorological observations. This could not be done using the RSs, since they lack from insufficient temporal and vertical resolution in the 0-10 m layer above the ground where the largest temperature gradients are observed. Furthermore, drones are able to study the influence of topography on the SBI structure and to measure extremely cold temperatures of air that can pool in topographic depressions. Finally, we demonstrated that drone measurements can determine the depth of unstable surface layers which are formed over sea ice during calm and clear conditions. All these unique capabilities by a drone can provide boundary layer meteorologists with a more realistic assessment of the processes that shape the temperature distribution in winter Arctic environments with important implications in the interpretation of regional variations in the skin–surface air temperature difference and the surface heat fluxes.

*Data availability.* Drone data are available from ABT upon request. Eureka A and Eureka C data are available online at the ECCC Historical Climate Data archive web page: https://climate.weather.gc.ca. FT data are available online at the NOAA Physical Science Laboratory web page: https://psl.noaa.gov/arctic/observatories/eureka/eureka_tower.html. Radiosonde data are available online at the web page of the Department of Atmospheric Science, University of Wyoming: http://weather.uwyo.edu/upperair/sounding.html. 0PAL data can be requested via the PEARL principal investigator or site manager. Details can be found at CANDAC web page: https://www.candac.ca.

*Author contributions.* ABT designed the study, developed the payload, conducted the measurement campaigns, and performed data quality control and analysis. GL designed the study and was a mentor of the NOAA FT. JRD was the principal investigator for the PEARL and facilitated the success of the project. All authors contributed to and commented on the paper.

*Competing interests.* The authors declare that they have no competing interests.

*Acknowledgements.* The authors would like to acknowledge the following groups and individuals for their technical support during field campaigns in Eureka: Canadian Arctic ACE/OSIRIS Validation Campaign project lead K. Walker, PEARL site manager P. Fogal, PEARL principal investigator K. Strong, CANDAC/PAHA operators: A. Hall, J. Gallagher, and P. McGovern; CANDAC/PAHA data manager Y. Tsehtik. Authors also thank the Eureka Weather Station staff for operation support. The authors acknowledge C. Cox, S. Morris, and T. Uttal (NOAA) for continuous support and operation of the FT in Eureka and for providing valuable data from it. ABT thanks M. Sauerberg

(AerialTech) for providing training on M210 RTK drone and J. Allen, M. Maurice and O. Stachowiak (ECCC) for providing details on Eureka meteorological and radiosonde data sets. ABT acknowledges A. Kräuchi, R. Philipona and K. Bärfuss for sharing their experience and fruitful conversations on the drones and their payloads. We thank E. McCullough for critically reading of the manuscript and valuable comments. Finally we appreciate the useful comments from M. Arshinov, J. Cassano, A. Leung and three anonymous reviewers.

PEARL research is supported by: the Canadian Foundation for Innovation; the Ontario Innovation Trust; the Ontario Ministry of Research and Innovation; the Nova Scotia Research and Innovation Trust; the Natural Sciences and Engineering Research Council (NSERC); the Canadian Foundation for Climate and Atmospheric Science; ECCC; Polar Continental Shelf Project; the Department of Indigenous and Northern Affairs Canada; and the Canadian Space Agency (CSA). This work was carried out at PEARL partially during 2017-2020 Canadian Arctic ACE/OSIRIS Validation Campaigns, which were funded by: CSA, ECCC, NSERC, and the Northern Scientific Training Program.

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
