# Peer review of "Drone Measurements of Surface-Based Winter Temperature Inversions in the High Arctic at Eureka"

_Atmospheric Measurement Techniques, 2020_

## Referee Comment (RC2)

Drone measurements of surface-based winter temperature inversions in the high Arctic at Eureka

by Tikhomirov et al.

The authors present measurements of temperature profiles obtained with quadrocopters in the high Arctic in winter under challenging environmental conditions. The description of the methodology is sound and of interest to a broad range of scientists. In particular the technical challenges that were encountered can be very valuable for other drone operators.

There are a few minor comments. My only major point is the suggestion to correct the measured profiles for time lag and take this into account for the analysis of lapse rate, which might be strongly influenced by the correction.

The article is clearly structured and well written.

Detailed comments:

Major points:

- The authors derive the lapse rate/strength of near-surface temperature inversions. However, as they point out, they do not correct the time lag of sensors. This can be seen by the disagreement of temperature profiles around the top of the profiles. As the quadrocopter goes up to the maximum flight altitude and subsequently down again, there is only a short time between the two measurements, and the temperature should be comparable. This obvious artefact induced by the measurement method/sensor characteristics should be corrected before deriving parameters like lapse rate and inversion strength. It would be nice to have two sub-plots of Fig. 5, one with the raw data like shown already, one with the corrected data. The large differences of temperature profiles for ascent and descent are clearly artefacts and not features. This is further underlines by the dependence of the differences on sensor position.
- You mention the influence on response time in l. 352. Please apply a correction, and compare also the correction to the literature.
- Further, it would be nice to embed the lapse rate observations more in the literature which describes such values.

Minor comments:

- The lapse rate is provided in °C/m and °C per km. This seems an unusual parameter to me. Mostly known in atmospheric science is the temperature change within 100 m (usually roughly within the range of plusminus 1°C). Another method would be to describe the temperature change within the 10 m altitude interval. Values like 300°C/km are difficult to understand at a first glance, and appear throughout the text.

- l. 18: is the heat flux through sea ice really called sensible heat flux? I would suggest to remove the "sensible". The term sensible heat flux usually refers to turbulent transport of heat from the ground into the atmosphere
- l. 25: if you mention the remote sensing techniques for satellite-based temperature measurements, please explain in more detail. In particular satellite based surface temperature measurements are strongly hampered by clouds. In any case I'm not aware of a satellite based method for deriving surface temperature inversions.
- l. 70/l. 76: I do not agree that fixed-wing aircraft are able to transport more payload than multirotor aircraft. If they have the same mass, let's say 25 kg, usually fixed-wing systems have a payload in the range of 5 kg plus batteries for an endurance of around 45 min flight time. Multirotor systems can handle easily up to 10 kg of payload, but with a typical endurance of 20 min. Please specify what exactly you mean here.
- l. 85: please be more specific about the advantages of unmanned systems compared to manned aircraft. In remote regions it may be easier to do measurements with a manned aircraft with longer endurance and without the need of access to the site. Further, manned aircraft usually allow to include more payload, which is clearly an advantage.
- please be specific about the usage of the terms "autonomous" and "automatic". Usually for drone operation, automatic refers to using an autopilot to fly along a given trajectory or way points. Autonomous means that you have a decision making instance on board, which can do tasks like detect and avoid. Not sure if you have this. In l. 309 it is mentioned that the "obstacle avoidance system" was disabled. This means that you were doing the flights in automatic mode.
- l. 233: Why did you choose the maximum flight altitude of 90 m? Was this an arbitrary decision? Why not 100 m? Were there restrictions of air space?
- l. 235, 240: repetition of favourable flight conditions
- l. 240: contradiction: you say that the relative humidity was 70%, and the air was very dry. Maybe you refer to absolute humidity of water vapour mixing ratio? Please specify.
- l. 248: add coordinates of measurement location
- l. 252: explain acronym GNSS when using it for the first time
- l. 268: refer to Fig. 4 for TS1
- l. 269/l. 324: explain FT earlier in the text
- l. 329: please explain more in detail how you investigate the influence of local topography
- l. 341: is the bias reproducible and can therefore be corrected?
- Fig.6/Fig 7: please use the same denominations for all flight legs and way points. What is "2-pass" (in the figure caption)? What is "–profile 2/3 passes" in the caption of Fig. 7? Does this correspond to waypoint p 3/5 in Fig. 6?
- Fig. 13: why is the style so different to the other figures? There are many more pixels / different line style.
- l. 414: maybe use the term "laser altimeter", if you only want to detect the ground return? Lidar may also refer to backscatter or wind lidar, which is not what you plan to use.

Suggestions for grammar/spelling:

- l. 17: above  sea ice
- l. 29: The WMO assesses global temperature
- l. 107: spent on  the development
- l. 132/137: same spelling: wheelbase or wheel base?
- l. 155: housed  in 25 mm….. tubes
- same style of date throughout the text, including year: 11 June 2019, also in tables, provide full date with year
- l. 232: The initial flight strategy
- l. 259: the drone performance
- l. 275: above  ground
- l. 334: wind speed
- l. 370: maintain the drone's altitude
- l. 391: this suggests that the local …
- l. 394: "created" instead of "creating"?
- l. 404: unclear: "and with 19:00 UTC Eureka C temperature". Please rephrase.
- l. 413: field operations (not filed)?
- l. 424: conducted with the M100 drone
- l. 437: "suggested" instead of "suggesting"

---

## Author Comment (AC3)

**Response to Prof. John Cassano (Referee #1) for AMT-2020-515.**

Dear Prof. Cassano, thank you for your interest in our work and valuable comments on the paper. On behalf of the co-authors, I am providing responses to your comments below. The line, page and figure numbers in {…} brackets correspond to the "latexdiff" version of the manuscript.

This paper describes the use of two different commercially available quadcopter drones for observing temperature inversions during the Arctic winter. Very strong temperature inversions, with temperature gradients of 100-300 K / km were observed immediately above the surface and matched in-situ observations from a 10 m tower and from radiosonde observations. This manuscript provides an excellent description of the technical challenges (related to quadcopter navigation and autopilot use) at high latitudes. The scientific results are limited to flights from a 7 day period in March 2020 although the data collected during these flights illustrate the very strong inversion conditions present at Eureka at the end of the winter. The main issue with the data shown in this manuscript are related to sensor lag, the impact of propellor downwash and sensor location on the quadcopter which result in non-negligible errors in the observed temperature. Despite this the results presented will be of interest to those hoping to use small drones to make meteorological measurements and I recommend that this manuscript be accepted for publication. Below I offer a few minor comments.

Specific comments

In the paragraph starting on line 65 it would be good to discuss some of the smaller fixed wing RPAS used for polar research as these are similar in terms of ease of deployment and payload capacity as multi rotor drones. Small Unmanned Meteorological Observer (SUMO)

Reuder et al. (2009)

Reuder et al. (2012)

Cassano (2014)

Jonassen et al. (2015)

DataHawk2

Lawrence and Balsley (2013)

de Boer et al. (2018)

Lines {139 -151}.

Done. The manuscript has been updated with a discussion of the smaller fixed-wing RPAS and references to corresponding papers.

Note that Cassano (2014) also note issues with sensor lag for ascent vs descent temperature profiles similar to what you have found.

Lines {139 -151}.

Done. The manuscript has been updated with a short summary of the results and a reference to the paper.

Line 226: Did you measure the battery temperature during the flight? If not, how do you know the battery temperature during the flights?

Lines {200-201, 209-211}.

Done. The stock batteries of both DJI M100 and DJI M210 RTK drones are equipped with internal temperature sensors, which continuously monitor battery core temperatures during the flight. The temperature readings are displayed on the screen of the drone's remote controller and stored in the log files onboard the drone together with other telemetry data.

The paragraphs describing drone's specification and battery enclosure have been populated with information about the internal battery temperature sensors.

A discussion of the temperature measurement issues (sensor lag, downwash impacting observed temperature and temperature differences for the RTD located at different positions on the quadcopter) should be discussed in the conclusions. These are the scientific issues that will limit the usefulness of quadcopters for making scientifically useful measurements of the near surface temperature profile.

Lines {656-668, 693-697}.

Done. A discussion has been added to section 3.2.5. and to the conclusion.

While not necessary to cite you may be interested in mid-latitude observations of cold pools using a bicycle based temperature sensor described in Cassano (2014b)

Thank you for useful reference and idea. We will consider the "Weather Bike" approach for our future studies in Eureka. We find this approach valuable and relatively easy to implement. The bike can be replaced by a RC controlled vehicle, and it could be a good addition to our drone measurements to study micro meteorological conditions and their links with local topographic features.

Technical corrections

Line 13: Replace of with above in "60 m of the ground"

Done.

Line 49: Replace one with was in "than one measured in Antarctica"

Done.

Line 131: Please include link to the relevant DJI web pages here.

Done.

Line 316: Replace one with that in "similar to one observed"

Done.

References

Cassano, J.J.: Observations of atmospheric boundary layer temperature profiles with a small unmanned aerial vehicle. Antarctic Science, 26, 205-213, doi:10/1017/S0954102013000539, 2014

Cassano, J.J., 2014b: Weather bike: A bicycle-based weather station for observing local temperature variations. Bulletin of the American Meteorological Society. doi:10.1175/BAMS-D-13-00044.1.

de Boer, G. et al. 2018, A bird's-eye view: Development of an operational ARM unmanned aerial capability for atmospheric research in Arctic Alaska. Bulletin of the American Meteorological Society, doi:10.1175/BAMS-D-17-0156.1.

Jonassen, M.O., Tisler, P., Altstädter, B., Scholtz, A., Vihma, T., Lampert, A., König-Langlo, G., and Lüpkes, C.: Application of remotely piloted aircraft systems in observing the atmospheric boundary layer over Antarctic sea ice in winter, Polar Research, 34, 25651, DOI:10.3402/polar.v34.25651, 2015.

Lawrence, D.A. and Balsley, B.B.: High-resolution atmospheric sensing of multiple atmospheric variables using the DataHawk small airborne measurement system. Journal of Atmospheric and Oceanic Technology, 30, 2352-2366, doi:10.1175/JTECHD-12-00089.1, 2013.

Reuder, J., Brisset, P., Jonassen, M., Müller, M. and Mayer, S., The Small Unmanned Meteorological Observer SUMO: A new tool for atmospheric boundary layer research. Meteorologische Zeitschrift, 18, 2, 141-147, 2009

Reuder, J., Jonassen, M.O., and Olafsson, H.: The Small Unmanned Meteorological Observer SUMO: Recent developments and applications of a micro-UAS for atmospheric boundary layer research. Acta Geophysica, 60, 1454-1473, doi:10.2478/s11600-012-0042-8, 2012.

Done. The manuscript has been updated with suggested references.

---

## Author Comment (AC4)

**Response to Referee #2 for AMT-2020-515.**

Dear Referee #2, thank you for your interest in our work and detailed review of our paper. On behalf of the co-authors, I am providing responses to your comments below. The line, page and figure numbers in {…} brackets correspond to the "latexdiff" version of the manuscript.

The authors present measurements of temperature profiles obtained with quadrocopters in the high Arctic in winter under challenging environmental conditions. The description of the methodology is sound and of interest to a broad range of scientists. In particular the technical challenges that were encountered can be very valuable for other drone operators.

There are a few minor comments. My only major point is the suggestion to correct the measured profiles for time lag and take this into account for the analysis of lapse rate, which might be strongly influenced by the correction.

The article is clearly structured and well written.

Detailed comments:

Major points:

- The authors derive the lapse rate/strength of near-surface temperature inversions. However, as they point out, they do not correct the time lag of sensors. This can be seen by the disagreement of temperature profiles around the top of the profiles. As the quadrocopter goes up to the maximum flight altitude and subsequently down again, there is only a short time between the two measurements, and the temperature should be comparable. This obvious artefact induced by the measurement method/sensor characteristics should be corrected before deriving parameters like lapse rate and inversion strength. It would be nice to have two sub-plots of Fig. 5, one with the raw data like shown already, one with the corrected data. The large differences of temperature profiles for ascent and descent are clearly artefacts and not features. This is further underlines by the dependence of the differences on sensor position.

Lines {402-420, 525-565}.

Done. We have applied a time lag correction to our raw temperature profiles from the flux tower flights and fjord flights following the approach reported by Cassano (2014). However, as it can be seen from updated figures in the manuscript, it does not always result in closely comparable temperature profiles measured on accent and descent. A discussion describing other sources that could contribute to the difference in the profiles has been included in the manuscript. For example, according to the data from NOAA Flux Tower, temperature variations on a scale of ~1C per minute occur nearly continuously during periods of extremely stable boundary conditions in Eureka (see Figure {6} in the updated version of the manuscript). Such fluctuations are natural and their effect could dominate over other factors in the calm or light wind conditions.

Figure {5} has been updated following the reviewer's suggestion.

Additionally, a discussion describing the reasons of the biases in the temperature readings from the sensors located at different places on the drone airframe has been included in the manuscript for clarification.

- You mention the influence on response time in l. 352. Please apply a correction, and compare also the correction to the literature.

Lines {402-420, 525-565, 693-697}

Done. We have applied a time lag correction to our raw temperature profiles from the flux tower and fjord flights following the approach reported by Cassano (2014). Our optimal time lag was found to be between 2 and 3.3 s. A comparison of our time lag with the literature values has been included in the manuscript.

- Further, it would be nice to embed the lapse rate observations more in the literature which describes such values.

Lines {62-73, 120-129}.

Done. The introduction has been updated with some extra references to the lapse rate measurements using radiosondes, drones and radiometers.

Minor comments:

- The lapse rate is provided in °C/m and °C per km. This seems an unusual parameter to me. Mostly known in atmospheric science is the temperature change within 100 m (usually roughly within the range of plusminus 1°C). Another method would be to describe the temperature change within the 10 m altitude interval. Values like 300°C/km are difficult to understand at a first glance, and appear throughout the text.

Done. The units in the manuscript have been updated. The lapse rate values have been given using both C/100 m and C/km dimensions.

- l. 18: is the heat flux through sea ice really called sensible heat flux? I would suggest to remove the "sensible". The term sensible heat flux usually refers to turbulent transport of heat from the ground into the atmosphere

Done.

- l. 25: if you mention the remote sensing techniques for satellite-based temperature measurements, please explain in more detail. In particular satellite based surface temperature measurements are strongly hampered by clouds. In any case I'm not aware of a satellite based method for deriving surface temperature inversions.

Lines {68-73}.

We do not see many reasons to describe remote sensing techniques for satellite-based temperature measurements in details in this manuscript, since this is not the goal of the manuscript and the information can be found somewhere else. Main caveats affecting the quality of the surface air

temperature datasets derived from the satellite-based measurements have been mentioned and references to some key publications have been included in the text.

In regards to characterisation of the surface temperature inversion derived from the satellite-based measurements please see Boylan, P., Wang, J., Cohn, S. A., Hultberg, T., and August, T. (2016), Identification and intercomparison of surface-based inversions over Antarctica from IASI, ERA-Interim, and Concordiasi dropsonde data, J. Geophys. Res. Atmos., 121, 9089– 9104, doi:10.1002/2015JD024724 and references.

- l. 70/l. 76: I do not agree that fixed-wing aircraft are able to transport more payload than multirotor aircraft. If they have the same mass, let's say 25 kg, usually fixed-wing systems have a payload in the range of 5 kg plus batteries for an endurance of around 45 min flight time. Multirotor systems can handle easily up to 10 kg of payload, but with a typical endurance of 20 min. Please specify what exactly you mean here.

Lines {98-119}.

We agree with the reviewer in this matter. This part of the manuscript has been rearranged to make our statements clearer.

- l. 85: please be more specific about the advantages of unmanned systems compared to manned aircraft. In remote regions it may be easier to do measurements with a manned aircraft with longer endurance and without the need of access to the site. Further, manned aircraft usually allow to include more payload, which is clearly an advantage.

Lines {110-119}.

Done. Specific clarification has been added to the beginning of the paragraph.

- please be specific about the usage of the terms "autonomous" and "automatic". Usually for drone operation, automatic refers to using an autopilot to fly along a given trajectory or way points. Autonomous means that you have a decision making instance on board, which can do tasks like detect and avoid. Not sure if you have this. In l. 309 it is mentioned that the "obstacle avoidance system" was disabled. This means that you were doing the flights in automatic mode.

Done.

We flew the drones using the autopilot along a set of preprogrammed way-points with the obstacle avoidance system being disabled, i.e. in automatic mode. The terminology in the manuscript has been fixed accordingly.

- l. 233: Why did you choose the maximum flight altitude of 90 m? Was this an arbitrary decision? Why not 100 m? Were there restrictions of air space?

Lines {326-339}.

More details have been added to the manuscript.

All drone operations reported in the manuscript were performed within the framework of the research activities conducted at PEARL and in accordance with Canadian Aviation Regulations for RPAS. Special procedures were established for operations in the vicinity of Eureka Aerodrome. The initial flight strategy consisted of several automatic (using an autopilot) or manual flights per day at various locations within FTS and RTS in the line-of-sight conditions with periodic ascents and descents.

Before June 1, 2019, the flights were conducted under Special Operation Flight Certificate, which restricted the maximum flight altitude for the drones to 91 m (300 ft) above the ground level, the minimum visibility - to 4.8 km (3 statute miles) and the minimum celling - to 305 m (1000 ft) above the ground level.

After June 1, 2019, the flights were conducted according to the updated Part IX of the Canadian Aviation Regulations, in which the maximum flight altitude for basic operations was extended to 122 m (400 ft) above the ground level.

To comply with the updated air space regulations and to increase the number of temperature profiles measured per flight before the drone batteries are drained, in 2020 our maximum flight altitude was 100 m above the ground level.

As can be seen from the manuscript, final maximum flight altitude did not exceed 60 m a.g.l. for the Flux Tower and Fjord flights and 100 m a.g.l. for the Gully flights.

- l. 235, 240: repetition of favourable flight conditions

Done.

- l. 240: contradiction: you say that the relative humidity was 70%, and the air was very dry. Maybe you refer to absolute humidity of water vapour mixing ratio? Please specify.

Lines {345-349}.

Done. This part has been modified as follows: "Potential challenges associated with propeller icing and darkness during the operations did not occur. At below -30C ambient temperature and at ~70% relative humidity (corresponds to 354~ppmv water vapour mixing ratio) the air was very dry and we did not observe any indications of icing on the propellers nor on the drone airframe during the flights (for comparison, 70% relative humidity at 0C corresponds to 4257~ppmv water vapour mixing ratio)."

- l. 248: add coordinates of measurement location

Done.

- l. 252: explain acronym GNSS when using it for the first time

Done.

- l. 268: refer to Fig. 4 for TS1

Done.

- l. 269/l. 324: explain FT earlier in the text

Done. Explained in the "2.3 Site description" section of the manuscript.

- l. 329: please explain more in detail how you investigate the influence of local topography

Lines {577-596}.

To investigate how local topography could influence the SBI a set of flights were conducted at the East side of the runway and in the gully near by. All the details and the results are discussed in section "3.2.3 Gully versus runway flights: SBI and local topography" of the manuscript.

- l. 341: is the bias reproducible and can therefore be corrected?

Lines {531-569}.

Yes, the bias is reproducible and can be corrected. However, we would like to avoid putting the temperature profiles from all 3 sensors in each temperature profile figure. Otherwise, the figures will be busy with data and hard to interpret. Since, the pole RTD was found to be the most accurate sensor onboard our drone, only the results from this sensor have been shown further in the paper. The manuscript has been updated with a discussion related to this.

- Fig.6/Fig 7: please use the same denominations for all flight legs and way points. What is "2-pass" (in the figure caption)? What is "–profile 2/3 passes" in the caption of Fig. 7? Does this correspond to waypoint p 3/5 in Fig. 6?

Done. Please, see updated Figures {7-11}.

- Fig. 13: why is the style so different to the other figures? There are many more pixels / different line style.

Done. The style has been updated to match the other figures. Please, see Figure {14}.

- l. 414: maybe use the term "laser altimeter", if you only want to detect the ground return? Lidar may also refer to backscatter or wind lidar, which is not what you plan to use.

Done.

Suggestions for grammar/spelling:

- l. 17: above the sea ice

Done.

- l. 29: The WMO assesses global temperature

Done.

- l. 107: spent on a the development

Done.

- l. 132/137: same spelling: wheelbase or wheel base?

Done. Replaced with "wheelbase" everywhere in the manuscript.

- l. 155: housed a in 25 mm….. tubes

Lines {229-230}.

Changed as follows: "The modules with RTD elements are housed in a 25 mm diameter and 75 mm long PVC tubes for protection."

- same style of date throughout the text, including year: 11 June 2019, also in tables, provide full date with year

Done.

- l. 232: The initial flight strategy

Done.

- l. 259: the drone performance

Done.

- l. 275: above the ground

Done.

- l. 334: wind speeds

Done.

- l. 370: maintain the drone's altitude

Done.

- l. 391: this suggests that the local …

Done.

- l. 394: "created" instead of "creating"?

Done.

- l. 404: unclear: "and with 19:00 UTC Eureka C temperature". Please rephrase.

Lines {633-635}.

Rephrased as follows: "Temperatures measured by the drone at 12 m above the sea ice ~210 and ~414 from the shoreline (pins #6 and #7 in Figure {3}) agree within ±0.5 C with the temperatures measured at Eureka C site at 18:00, 19:00 and 20:00 UTC."

- l. 413: field operations (not filed)?

Done.

- l. 424: conducted with the M100 drone

Done.

- l. 437: "suggested" instead of "suggesting"

Done. Used "suggest" instead to match present tense tone of the Conclusion section.

References:

Cassano, J.J.: Observations of atmospheric boundary layer temperature profiles with a small unmanned aerial vehicle. Antarctic Science, 26, 205-213, doi:10/1017/S0954102013000539, 2014

---

## Author Comment (AC5)

**Response to Referee #3 for AMT-2020-515.**

Dear Referee #3, thank you for your interest in our work and detailed review of our paper. On behalf of the co-authors, I am providing responses to your comments below. The line, page and figure numbers in {…} brackets correspond to the "latexdiff" version of the manuscript.

The authors present field deployments testing two types of quad-copters in the harsh conditions of the high-Arctic in winter. The technical description and challenges are relevant and clearly articulated. In general, the work presented here seems very useful for the establishment of high Arctic measurements by drones. Nevertheless, some of the technical issues discovered were only stated but no possible amendments were suggested/applied. Also, the presentation of the results needs some augmentation, especially when comparing to local measurements. More statistical analysis and discussion might be warranted, especially since the current study findings show much deeper inversions than previously suggested for the Arctic (versus the Antarctic).

Minor comments:

Line 17 in the abstract: agrees well with the one (one comparison seems odd)

Lines {19-20}

Done. "The inversion lapse rate agrees well with the results obtained from the radiosonde temperature measurements."

Line 56, positively correlated with what?

Lines {83-84}

Corrected as follows: "They found the strength, occurrence frequency and depth of the SBI are larger in winter and fall than in summer and spring and are positively correlated between each other, both spatially and temporally."

Line 99-100, sentence too long; consider parsing

Lines {164-167}

Done. Parsed as suggested.

Line 123, on SBI shaping

Done.

Line 144, specification of 0.1 m

Lines {216-218}

Done. Corrected as follows: "According to specification a hovering accuracy of 0.1 m in both vertical and horizontal directions can be reached by utilizing the drone together with DJI D-RTK ground system kit (RTK mode)."

Line 168, the results showed good agreement (can you please specify the correlation coefficient? Number of data points?)

Lines {240-243}

Done. The BMP280 altimeter was verified by comparing its pressure readings to simultaneous measurements taken with a Vaisala WXT-520 weather transmitter within the pressure range between 92.5 and 100.2 kPa. Both sensors were placed in a truck and collected data at 10 Hz sampling rate while the truck was driven from the PEARL Ridge Laboratory (610 m.a.s.l.) down to the Eureka Weather Station (10 m.a.s.l.). Total number of data points is equal to 14791. Pearson's correlation coefficient is equal to 0.99999. Linear fit R-Square coefficient is equal to 0.99997. Plots representing the results of validation of BMP280 pressure sensor against Vaisala WXT-520 weather transmitter are shown below.

[Figure]

[Figure]

[Figure]

Figure 5 and its discussion: the results shown are for one short profile. I would expect seeing some additional statistics and comparisons with the FT and radiosonde data during the campaign duration, to support or decline the proposed biases and/or agreements.

We are not able to provide additional statistics and comparisons of drone data with the FT data for 2020 for now. The reasons for this are as follows. Unfortunately, 2020 FT data are still not available. This is due to a failure in the FT data transfer system which is not possible to fix till somebody of our staff travels to Eureka. Since the staff is still under travel restrictions due to COVID-19 pandemic and we are still waiting for an approval from Canadian government to travel to the site, there is no timeline for the trip yet.

Some statistics could be retrieved from the measurements made in 2017-2019. However, during that period we were developing the drone temperature measurement technique. The flights were sporadic in time with a large variation of flight parameters and several drone failures.

In regards to the drone vs radiosonde comparisons, it is not possible to provide valid drone vs radiosonde comparisons for the layer <10 m above the ground, since radiosondes can not resolve this layer well. Within 10-60 m above the ground our drone lapse rates agree well with the lapse rates obtained from the Eureka RSs, launched within 4 hours after the drone flights and are close to the results of multi-year studies conducted by Bradley et al. (1993) and by Walden et al. (1996) based on Eureka RS data covering 1967-1990.

While the drone and the RS lapse rates agree, spatial separation between ECCC WS and RTS, temporal separation between the drone flights and radiosonde launches and up to several degrees natural temperature variations on a time scale ranging from 1 minute to several hours make the results of the comparisons of drone vs RS absolute temperatures for 10-100 m altitudes challenging. To address this, we plan to conduct drone temperature measurements simultaneously with the radiosonde launches in the future as soon as we can travel to Eureka. Other option is to install RS onboard the drone and conduct simultaneous temperature measurements using drone and RS temperature sensors.

Also, the temperature measurements show lapse rates much higher than previously observed in the Arctic region and this also calls for some additional discussion.

Lines {682-686, 701-707}

Done. The conclusion has been updated with additional discussion and comparisons.

Lines 355-357, the temperature profile differences between the passes in Figure 7 are not always "slight"; there are some variations of 1 degree (subplot c), which warrants more attention. Is there a reference instrument that can attest to the change in ambient conditions over this time?

Lines {548-565}

There is no closely located reference instrument that can attest to the change in ambient conditions over the time of our flights, except the temperature sensors installed at the NOAA Flux Tower (FT).

Unfortunately, 2020 FT temperature data are still not available.

According to the FT data from 2017 and early years, temperature variations on a scale of ~1C per minute occur nearly continuously during periods of extremely stable boundary conditions in Eureka (see Figure {6} in the updated version of the manuscript). Such fluctuations are natural. They could be the reason of the differences in the drone temperature profiles. A discussion describing the temperature variations based on 2017 FT data has been included in the manuscript (see section 3.1.2 and Figures {5} and {6}).

Eureka A (the closest alternative weather station to the RTS) and Eureka Climate (the closest weather station to the FTS) temperature data are available, but only at 1 hour resolution.

Figure 9 caption: probably course and not coarse

Done.

Line 370, not able to maintain

Lines {584-585}

Done. Rephrased as follows: "During this flight the RTK system experienced a number of intermittent failures and the drone was not able to maintain its altitude properly."

Figure 10, it is interesting to see the better agreement for the FT flights; is this a coincidence or the result of the location/specific conditions? Were additional comparisons were made with the FT?

Page {35}.

Figure {11} shows temperature profiles measured during M210 RTK flux tower flight along WP1-WP7 way-points on 9 March 2020. Temperature profiles from the RS launched from the ECCC WS at 23:15 UTC and corrected for the altitude difference between the ECCC WS and the RTS elevations are depicted in the figure together with 18:00, 19:00 and 20:00 UTC Eureka A temperatures.

A comparison of the FT temperature data for 28 February 2017 with simultaneously measured drone temperature profiles and 11:15 and 23:15 UTC RS profiles (see Figure {5}) suggests that a good

agreement between the drone and RS temperature profiles in Figure {11} could be coincident. Most of the drone temperature profiles measured at the RTS in 2020 show positive bias in comparison with the RS profiles (see Figures {8-11}). Horizontal and vertical separation between the RS launch site and the RTS, time difference between the measurements and natural temperature variations makes the results of the comparisons of drone vs RS absolute temperatures challenging. RS measurements conducted simultaneously with the drone and FT measurements would clarify whether this a coincidence or the result of the location/specific conditions.

Please, all see the answers to the comments above and below, since they are related to each other.

Fig.11-12, I think there is room for further discussion of the results, especially on the different profile shapes on the different days and their similarity (dissimilarity) on the different days in the gully vs. the runway. For example, profile shapes in Fig. 12 seem similar but shifted between the gully and runway, while Fig. 11 shows more similar trends. More discussion on the conditions and why we see these results.

As it was mentioned above, 2020 temperature data from our reference instrument – the FT, are still not available. We would like to avoid further discussion of the conditions that could cause the similarity (dissimilarity) in the gully vs the runway temperature profiles without these data. Therefore, we would like to present the gully vs the runway results as is for now and get back to detailed analysis as soon as the FT data become available. Although, two potential reasons that could cause the similarity (dissimilarity) in the profiles have been briefly highlighted at the end of section 3.2.3.

Line 413: Field instead of Filed

Done.

Lines 413-415, while IR sensor and a Lidar can be a valuable addition, these are much heavier instruments and more complicated installations, so mentioning their addition as a side-note might not be appropriate without further scrutinization. Maybe state the expected challenges from such add-ons.

Lines {649-654}.

An IR sensor is a small instrument. There are many lightweight "turn-key" solutions and development boards (<50 g) with a variety of communication interfaces (USB, I2C, SPI) available commercially.

By the Lidar we meant Laser Altimeter/Rangefinder. The manuscript has been corrected accordingly. Lightweight (~100-200 g) altimeters with an operation range of 100-200 m are also commercially available.

We do not expect many challenges in implementing such add-ons.

---

## Author Comment (AC6)

**Response to Andrew Leung for AMT-2020-515.**

Dear Dr. Leung, thank you for your interest in our work and valuable comments on the paper. On behalf of the co-authors, I am providing responses to your comments below. The line, page and figure numbers in {…} brackets correspond to the "latexdiff" version of the manuscript.

There is very few research conducted in High Arctic, due to the high cost of travel, short field season and being difficult to access. This particular paper seeks to improve the measurement technologies in this area. I have some comments for this preprint.

In line 43, I understand the need to express units in SI units. However, it's preferable to express the inversion lapse rate in metre or 100 metres. Same for lapse rates in lines 48 and 50. The authors already express lapse rate in metre on lines 275 and 276

Done. The units in the manuscript have been updated. The lapse rate values have been given using both C/100 m and C/km dimensions.

In line 179, CYEU should be written as CYEU. C-YEU would be an aircraft registration number, not an airport designation code

Done.

In line 185, the Eureka Climate station name should be spelled out in full at least in the first instance, not abbreviated as "Eureka C"

Done.

From line 185 to 188, when the manuscript is referring to Nav Canada's Eureka A station, it is unclear if the authors are referring to the data from the automatic platform (which runs 24/7) or the staffed platform (which runs 22 hours a day during their study period, with no staff present for 04:00 and 05:00 UTC observations). Both the automatic station share the same instruments (e.g. temperature, wind, pressure) but staffed site also provide hourly weather conditions and visibility, and 6-hr precip amounts. If authors used the staffed site, then the statement "both stations provide hourly weather reports" is not accurate. The manuscript should clarify which platform they used in their study.

Lines {259-270}

To characterize meteorological conditions at the sites we have used data on temperature, RH, wind, pressure from the Eureka A and Eureka Climate (Eureka C) automatic platforms as well as data on hourly weather conditions and visibility provided by ECCC staff at Eureka Weather Station. The manuscript has been updated with proper clarification.

In lines 208 to 209, do you have the source for the ice surveys? These information should be available on Canadian Ice Service website

Lines {298, 626}

Done. The paragraphs have been rearranged and a reference to the Canadian Ice Service website has been included.

Just a thought, since the authors built insulation to protect the battery against extreme cold as stated in line 225, was the battery temperature measured or tracked during the flight?

Lines {200-201, 209-211}.

Done. The stock batteries of both DJI M100 and DJI M210 RTK drones are equipped with internal temperature sensors, which continuously monitor battery core temperatures during the flight. The temperature readings are displayed on the screen of the drone's remote controller and stored in the log files onboard the drone together with other telemetry data.

The paragraphs describing drone's specification and battery enclosure have been populated with information about the internal battery temperature sensors.

It would be ideal to clarify in line 236 that the pilot is the drone pilot

Done.

Figures 3, 4 and 6 are labelled as products copyrighted by Google Earth. It is not compatible with the Creative Commons (CC-BY-4.0) license. If alternative imageries (e.g. NASA) in public domain are available, they should be used instead

According to the AMT submission rules (https://www.atmospheric-measurement-techniques.net/submission.html) reproduction and reuse of maps and aerials which are not compatible with CC-BY licence (such as provided by Google Maps, Google Earth) is acceptable as long as the content includes "the required copyright and distribution licence statements of the map provider".

According to Google Guidelines (https://www.google.com/intl/en-GB_ALL/permissions/geoguidelines/ and https://about.google/brand-resource-center/products-and-services/geo-guidelines/) "Google Earth or Earth Studio can be used for purposes such as **research**, education, film and nonprofit use **without needing permission**" and if all the content created from Google Earth or Earth Studio is properly attributed.

Figures {3, 4 and 7} have been updated to include all required information accordingly.

The imageries of Eureka provided by NASA and available at https://worldview.earthdata.nasa.gov/, unfortunately, have resolution, unacceptable for our applications.

USGS maps available at https://earthexplorer.usgs.gov/ have resolution comparable to Google Earth imageries, but they are "not for purchase or for download" and "to be used as a guide for reference and search purposes only" according to USGS statement.

In Figures 3 and 4, it is unclear to me what those alphabets on the map pins represent. For example, I see two pins labelled as "E". Their meanings were not explained in figure captions.

Done. The manuscript, Figures {3, 4 and 7} and their captions have been updated to provide details on the meaning of the pins, way-points, drone flight trajectories and the locations of various facilities in Eureka.

---

## Referee Report (RR1)

**General comments:**

This is a useful research article within the scope of the journal. The potential for using drones to measure the characteristics of temperature inversion in challenging conditions has been demonstrated, and the critical points of this task have been discussed in detail. A very good review of the references is given.

However, the wrong choice of drone and temperature measurement system limited the quality of this research. Also, the altitude of all flights was below 60 meters, and some even less than 30 meters, which is a very limited altitude range for drone measurements.

**Specific comments:**

Line 13: Why did you limit the flight altitude to less than 60 meters? You had the ability to measure a much wider range of altitudes above ground level.

Line 185: Why did you choose a commercial drone over an open-source solution? Such closed product does not allow adjustments of the drone for this purpose. You could solve the barometer problems by using a laser altimeter, which is very easy to implement in the case of an open-source platform. You could also try different magnetic field sensors and GNSS receivers with interchangeable antennas, etc. Bad drone selection has made solving the problems arising from challenging atmospheric conditions impossible.

Line 216: The declared accuracy of the temperature measurement system is poor for this purpose. You did not calibrate the temperature measurement system before the measurement campaign, but in the last phase of the research you did the validation and stated that the temperature measurement system has shortcomings.

Line 217: placing the sensor in a protective PVC tube has an effect on the response time of the sensor. You should have measured the sensor response time, at least in the laboratory.

Line 367: You state the response time of the sensor from the manufacturer's specification, and you said earlier that you protected the sensor with a PVC tube. Based on Figure 5 showing the thick hysteresis in the temperature graphs, I suspect that the actual response time of your sensor is much longer than the value from the specifications.

Line 410: You had to do laboratory tests and calibration of the temperature measurement system before the field measurement campaign, not at this stage of the research.

Line 423: You (correctly) state here that a closed commercial drone, a "black box", prevents you to solve technical problems.

In Chapter 3.2.5, the authors realize that poor equipment selection has limited their research.

Nevertheless, despite these shortcomings, this paper contains useful information and results and complements previous research of the phenomenon of temperature inversion using drones.

---

## Author Response (AR2)

**Response to Referee #4 for AMT-2020-515.**

Dear Referee #4, thank you for your interest in our work and detailed review of our paper. On behalf of the co-authors, I am providing responses to your comments below. The line, page, and figure numbers in {…} brackets correspond to the "latexdiff" version of the manuscript. Corrections are marked in blue color in the "latexdiff" version of the manuscript.

**General comments:**

This is a useful research article within the scope of the journal. The potential for using drones to measure the characteristics of temperature inversion in challenging conditions has been demonstrated, and the critical points of this task have been discussed in detail. A very good review of the references is given.

However, the wrong choice of drone and temperature measurement system limited the quality of this research. Also, the altitude of all flights was below 60 meters, and some even less than 30 meters, which is a very limited altitude range for drone measurements.

**Specific comments:**

1. Line 13: Why did you limit the flight altitude to less than 60 meters? You had the ability to measure a much wider range of altitudes above ground level.

Please, see lines {13, 167, 311-317, 319, 582, 590-599, 604, 607} and Figures {12 and 13}.

We realized that our statement about 60 m maximum flight altitude requires a correction.

For the flux tower flights, drone altitudes did not exceed a maximum of 60 m above the ground. For the gully versus runway flights, they did not exceed 75 m and 70 m above the gully and runway surface levels correspondingly. For the fjord flights, drone altitudes did not exceed a maximum of 55 m above the ice surface.

There are three reasons such altitudes were used:

1. In our SBI measurements we focused our interest on 0-100m layer where strong temperature gradients were observed.
2. Since the flights were conducted at low ascent/descent speeds (0.1-0.7 m/s) our priority was to measure as many temperature profiles per flight as possible before the drone batteries were drained. Due to this, in most of the flux tower flights, drone altitudes did not exceed a maximum of 40 m above the ground.
3. As we mentioned in the manuscript (see lines {311-317}), to comply with the Canadian Aviation Regulations, during our 2017 and 2020 flight campaigns drone maximum altitudes were kept within 91 m (300 ft) and 122 m (400 ft) above the ground level, correspondingly.

The numbers in the manuscript as well as Figures {12 and 13} have been corrected accordingly. Also, a clarification has been added to the conclusion of the manuscript.

2. Line 185: Why did you choose a commercial drone over an open-source solution? Such closed product does not allow adjustments of the drone for this purpose. You could solve the barometer problems by using a laser altimeter, which is very easy to implement in the case of an open-source platform. You could also try different magnetic field sensors and GNSS receivers with interchangeable antennas, etc. Bad drone selection has made solving the problems arising from challenging atmospheric conditions impossible.

Please, see lines {163-168, 566-589}.

Among the goals of the paper was to demonstrate and evaluate the feasibility of drone operations at 80N. We were driven by the idea of using a commercial "turn-key" drone solution for our application. Keeping this in mind, the plan was to evaluate and learn whether an "off-the-shelf" rotary-wing drone can be economic, robust, and reliable in the High Arctic environment, so the time and efforts spent on the development of a custom system can be saved. This has been clearly stated in the Introduction.

While we had issues with our M100 drone, the M210 RTK drone equipped with RTK navigation system allowed us to conduct automatic flights in RTK mode.

As we stated in section "3.2.5 Lessons learned and future prospects", in the future, our payload will be improved by an installation of a laser altimeter which would provide information about drone's altitude above the ground with a precision better than barometric altimeter. The altimeter also can be used to track fine scale topography of the surface during the flight.

3. Line 216: The declared accuracy of the temperature measurement system is poor for this purpose. You did not calibrate the temperature measurement system before the measurement campaign, but in the last phase of the research you did the validation and stated that the temperature measurement system has shortcomings.

Please, see lines {423-426}.

Unfortunately, during our studies we did not have access to a test chamber to calibrate our sensors.

The most significant uncertainty in our temperature measurements is associated with the tolerance class of Omega 1PT100KN1510 RTDs (Class B, W0.3, total accuracy $\Delta T = \pm(0.30 + 0.0050 \,|T|)$, where T is temperature in [C]). For 0, -40 and -50 C temperatures this results in ±0.30, ±0.50 and ±0.55C measurement accuracy, correspondingly. High linearity and stability of platinum RTDs together with the results of validation of the sensors in the melting ice and application of bias correction (biases did not exceed -0.003±0.013, 0.25±0.02 and 0.30±0.02 for the pole, top and rotor 3 RTDs) allow us to conclude that the accuracy of our temperature measurements is ~0.3 C for -50 to -40C temperature range.

This value is equal to the required instrument measurement uncertainty recommended by WMO for below -40 air temperatures (Guide to Instruments and Methods of Observation, Volume I – Measurement of Meteorological Variables, 2018 edition, page 24: https://library.wmo.int/doc_num.php?explnum_id=10616).

The manuscript has been updated accordingly.

4. Line 217: placing the sensor in a protective PVC tube has an effect on the response time of the sensor. You should have measured the sensor response time, at least in the laboratory.

Please, see lines {412-414}

We understand that. For this reason, in 2020 the pole PVC tube was equipped a with fan, which provided aspiration for the pole RTD element to improve the response time.

5. Line 367: You state the response time of the sensor from the manufacturer's specification, and you said earlier that you protected the sensor with a PVC tube. Based on Figure 5 showing the thick hysteresis in the temperature graphs, I suspect that the actual response time of your sensor is much longer than the value from the specifications.

Please, see lines {412-414, 445-447, 615-619} and Figures {5, 9-11}

Figure {5} represents the results of the measurements conducted in 2017. At that time neither of the sensors were forcibly aspirated.

In 2020 we introduced aspiration of the pole RTD element by a fan. Additionally, we decreased the drone's speed of ascent/descent from 1-2.8 m/s down to 0.1-0.7 m/s. This allowed us to decrease the effect of the response time on the measurements and, hence, minimize the hysteresis (please, see Figures {9-11}.

6. Line 410: You had to do laboratory tests and calibration of the temperature measurement system before the field measurement campaign, not at this stage of the research.

Please, see lines {415-426}.

Please, also see our response to the comment #3 above.

During our studies we did not have access to a test chamber to calibrate our temperature sensors. Due to that, we conducted pre- and post-flight validation of the sensors.

7. Line 423: You (correctly) state here that a closed commercial drone, a "black box", prevents you to solve technical problems.

Please, see our response to the comment #2 above.

8. In Chapter 3.2.5, the authors realize that poor equipment selection has limited their research.

Please, see our response to the comment #2 above.

9. Nevertheless, despite these shortcomings, this paper contains useful information and results and complements previous research of the phenomenon of temperature inversion using drones.

We appreciate the feedback provided by the referee very much.

We would like to include the following additional references to the manuscript:

1. Kral, S.T.; Reuder, J.; Vihma, T.; Suomi, I.; O'Connor, E.; Kouznetsov, R.; Wrenger, B.; Rautenberg, A.; Urbancic, G.; Jonassen, M.O.; Båserud, L.; Maronga, B.; Mayer, S.; Lorenz, T.; Holtslag, A.A.M.; Steeneveld, G.-J.; Seidl, A.; Müller, M.; Lindenberg, C.; Langohr, C.; Voss, H.; Bange, J.; Hundhausen, M.; Hilsheimer, P.; Schygulla, M. Innovative Strategies for Observations in the Arctic Atmospheric Boundary Layer (ISOBAR)—The Hailuoto 2017 Campaign. Atmosphere 2018, 9, 268. https://doi.org/10.3390/atmos9070268

2. Varentsov, M.; Stepanenko, V.; Repina, I.; Artamonov, A.; Bogomolov, V.; Kuksova, N.; Marchuk, E.; Pashkin, A.; Varentsov, A. Balloons and Quadcopters: Intercomparison of Two Low-Cost Wind Profiling Methods. Atmosphere 2021, 12, 380. https://doi.org/10.3390/atmos12030380

3. Wenta, M., Brus, D., Doulgeris, K., Vakkari, V., and Herman, A.: Winter atmospheric boundary layer observations over sea ice in the coastal zone of the Bay of Bothnia (Baltic Sea), Earth Syst. Sci. Data, 13, 33–42, https://doi.org/10.5194/essd-13-33-2021, 2021.

4. Barbieri, L.; Kral, S.T.; Bailey, S.C.C.; Frazier, A.E.; Jacob, J.D.; Reuder, J.; Brus, D.; Chilson, P.B.; Crick, C.; Detweiler, C.; Doddi, A.; Elston, J.; Foroutan, H.; González-Rocha, J.; Greene, B.R.; Guzman, M.I.; Houston, A.L.; Islam, A.; Kemppinen, O.; Lawrence, D.; Pillar-Little, E.A.; Ross, S.D.; Sama, M.P.; Schmale, D.G.; Schuyler, T.J.; Shankar, A.; Smith, S.W.; Waugh, S.; Dixon, C.; Borenstein, S.; de Boer, G. Intercomparison of Small Unmanned Aircraft System (sUAS) Measurements for Atmospheric Science during the LAPSE-RATE Campaign. Sensors 2019, 19, 2179. https://doi.org/10.3390/s19092179

5. Segales, A. R., Greene, B. R., Bell, T. M., Doyle, W., Martin, J. J., Pillar-Little, E. A., and Chilson, P. B.: The CopterSonde: an insight into the development of a smart unmanned aircraft system for atmospheric boundary layer research, Atmos. Meas. Tech., 13, 2833–2848, https://doi.org/10.5194/amt-13-2833-2020, 2020.

The following figures have been also updated:

Figures {6, 12, 13, 14}